# Highly efficient 5′ capping of mitochondrial RNA with NAD+ and NADH by yeast and human mitochondrial RNA polymerase

Jeremy G Bird[1,2†], Urmimala Basu[3,4†], David Kuster[1,2,5], Aparna Ramachandran[3], Ewa Grudzien-Nogalska[6], Atif Towheed[7], Douglas C Wallace[7,8], Megerditch Kiledjian[6], Dmitry Temiakov[9], Smita S Patel[3]*, Richard H Ebright[2]*, Bryce E Nickels[1]*

[1]Department of Genetics and Waksman Institute, Rutgers University, United States; [2]Department of Chemistry and Waksman Institute, Rutgers University, United States; [3]Department of Biochemistry and Molecular Biology, Robert Wood Johnson Medical School, Rutgers University, United States; [4]Biochemistry PhD Program, School of Graduate Studies, Rutgers University, United States; [5]Biochemistry Center Heidelberg, Heidelberg University, Germany; [6]Department of Cell Biology and Neuroscience, Rutgers University, United States; [7]Center for Mitochondrial and Epigenomic Medicine, The Children's Hospital of Philadelphia, United States; [8]Department of Pediatrics, Division of Human Genetics, The Children's Hospital of Philadelphia, Perelman School of Medicine, United States; [9]Department of Biochemistry and Molecular Biology, Sidney Kimmel Cancer Center, Thomas Jefferson University, United States

*For correspondence:
patelss@rwjms.rutgers.edu (SSP);
ebright@waksman.rutgers.edu
(RHE);
bnickels@waksman.rutgers.edu
(BEN)

†These authors contributed
equally to this work

Competing interests: The
authors declare that no
competing interests exist.

Reviewing editor: Alan G
Hinnebusch, Eunice Kennedy
Shriver National Institute of Child
Health and Human
Development, United States

**Abstract** Bacterial and eukaryotic nuclear RNA polymerases (RNAPs) cap RNA with the oxidized and reduced forms of the metabolic effector nicotinamide adenine dinucleotide, NAD+ and NADH, using NAD+ and NADH as non-canonical initiating nucleotides for transcription initiation. Here, we show that mitochondrial RNAPs (mtRNAPs) cap RNA with NAD+ and NADH, and do so more efficiently than nuclear RNAPs. Direct quantitation of NAD+- and NADH-capped RNA demonstrates remarkably high levels of capping in vivo: up to ~60% NAD+ and NADH capping of yeast mitochondrial transcripts, and up to ~15% NAD+ capping of human mitochondrial transcripts. The capping efficiency is determined by promoter sequence at, and upstream of, the transcription start site and, in yeast and human cells, by intracellular NAD+ and NADH levels. Our findings indicate mtRNAPs serve as both sensors and actuators in coupling cellular metabolism to mitochondrial transcriptional outputs, sensing NAD+ and NADH levels and adjusting transcriptional outputs accordingly.
DOI: https://doi.org/10.7554/eLife.42179.001

## Introduction

Chemical modifications of the RNA 5′-end provide a layer of 'epitranscriptomic' regulation, influencing RNA fate, including stability, processing, localization, and translation efficiency (*Höfer and Jäschke, 2018*; *Jäschke et al., 2016*; *Ramanathan et al., 2016*; *Shuman, 2015*). One well-characterized RNA 5′-end modification is the 'cap' comprising 7-methylguanylate (m7G) added to many eukaryotic messenger RNAs (*Furuichi and Shatkin, 2000*; *Shatkin, 1976*; *Shuman, 1995*; *Wei et al.,*

*1975*). Recently, a new RNA 5'-end cap comprising the metabolic effector nicotinamide adenine dinucleotide (NAD) has been shown to be added to certain RNAs isolated from bacterial, yeast, and human cells (*Cahová et al., 2015*; *Chen et al., 2009*; *Frindert et al., 2018*; *Jiao et al., 2017*; *Walters et al., 2017*).

In contrast to a $m^7G$ cap, which is added to nascent RNA by a capping complex that associates with eukaryotic RNA polymerase II (RNAP II) (*Ghosh and Lima, 2010*; *Martinez-Rucobo et al., 2015*; *Shuman, 1995*; *Shuman, 2001*; *Shuman, 2015*), an NAD cap is added by RNAP itself during transcription initiation, by serving as a non-canonical initiating nucleotide (NCIN) (*Bird et al., 2016*) (reviewed in *Barvík et al., 2017*; *Julius et al., 2018*; *Vasilyev et al., 2018*). NCIN-mediated NAD capping has been demonstrated for bacterial RNAP (*Bird et al., 2016*; *Frindert et al., 2018*; *Julius and Yuzenkova, 2017*; *Vvedenskaya et al., 2018*) and eukaryotic RNAP II (*Bird et al., 2016*). Thus, whereas $m^7G$ capping occurs after transcription initiation, on formation of the ~20th phosphodiester bond, and occurs only in organisms harboring specialized capping complexes, NAD capping occurs in transcription initiation, on formation of the first phosphodiester bond, and because it is performed by RNAP itself, is likely to occur in most, if not all, organisms.

NAD exists in oxidized and reduced forms: $NAD^+$ and NADH, respectively (*Figure 1A*). Capping with $NAD^+$ has been demonstrated both in vitro and in vivo (*Bird et al., 2016*; *Frindert et al., 2018*; *Julius and Yuzenkova, 2017*; *Vvedenskaya et al., 2018*). Capping with NADH has been demonstrated in vitro (*Bird et al., 2016*; *Julius and Yuzenkova, 2017*).

Jäschke and co-workers developed a method that combines click-chemistry-mediated covalent capture and high-throughput sequencing, 'NAD captureSeq,' to detect $NAD^+$-capped RNA (*Cahová et al., 2015*; *Winz et al., 2017*). Jäschke and co-workers used this method to identify $NAD^+$-capped RNAs in bacterial cells (*Escherichia coli* and *Bacillus subtilis*; *Cahová et al., 2015*; *Frindert et al., 2018*). Parker, Kiledjian, and co-workers used the same method to identify $NAD^+$-capped RNAs in eukaryotic cells (*Saccharomyces cerevisiae* and human cell line HEK293T; *Jiao et al., 2017*; *Walters et al., 2017*). Notably, the identified *Saccharomyces cerevisiae* $NAD^+$-capped RNAs included not only RNAs produced by nuclear RNAPs, but also RNAs produced by mitochondrial RNAP (mtRNAP). The eukaryotic nuclear RNAPs–RNAP I, II, and III–are multi-subunit RNAPs closely related in sequence and structure to bacterial RNAP (*Cramer, 2002*; *Darst, 2001*; *Ebright, 2000*; *Werner and Grohmann, 2011*); in contrast, mtRNAPs are single-subunit RNAPs that are unrelated in sequence and structure to multi-subunit RNAPs and, instead, are related to DNA polymerases, reverse transcriptases, and DNA-dependent RNAPs from T7-like bacteriophages (*Cermakian et al., 1996*; *Cheetham and Steitz, 2000*; *Hillen et al., 2018*; *Masters et al., 1987*; *McAllister and Raskin, 1993*; *Ringel et al., 2011*; *Sousa, 1996*).

The identification of $NAD^+$-capped mitochondrial RNAs in *S. cerevisiae* raises the question of whether eukaryotic single-subunit mtRNAPs–like the structurally unrelated bacterial and eukaryotic nuclear multi-subunit RNAPs–can perform NCIN-mediated capping. A recent review discussed evidence supporting the hypothesis that human mtRNAP can perform NCIN capping (*Julius et al., 2018*). Here, we show that single-subunit *S. cerevisiae* mtRNAP and human mtRNAP perform NCIN-mediated capping with $NAD^+$ and NADH in vitro, and do so substantially more efficiently than bacterial and eukaryotic multi-subunit RNAPs. Further, we show that capping efficiency is determined by promoter sequence, we demonstrate very high levels of $NAD^+$ and NADH capping–up to ~60% of mitochondrial transcripts in vivo, and we demonstrate that the extents of capping in vivo, and distributions of $NAD^+$ capping vs. NADH capping in vivo are influenced by intracellular levels of $NAD^+$ and NADH.

## Results

### *S. cerevisiae* and human mtRNAPs cap RNA with $NAD^+$ and NADH in vitro

To assess whether mtRNAP can cap RNA with $NAD^+$ and NADH, we performed in vitro transcription experiments (*Figure 1* and *Figure 1—figure supplement 1*). We analyzed *S. cerevisiae* mtRNAP and a DNA template carrying the *S. cerevisiae* mitochondrial 21S promoter (*Deshpande and Patel, 2014*), and, in parallel, human mtRNAP and a DNA template containing a derivative of the human mitochondrial light-strand promoter, $LSP_{AGU}$ (*Sologub et al., 2009*) (*Figure 1C–D*, top). We

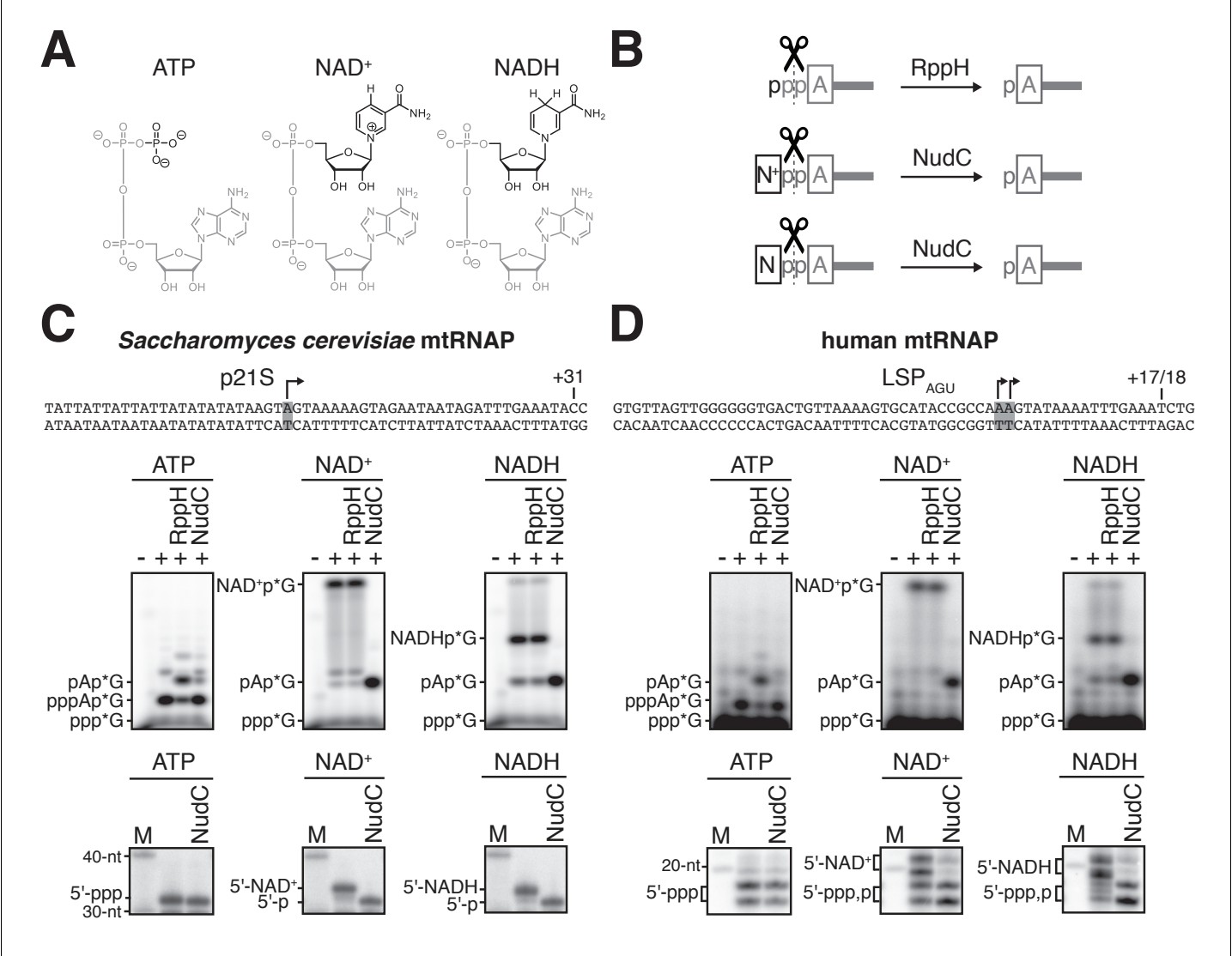

**Figure 1.** *S. cerevisiae* and human mtRNAPs cap RNA with NAD$^+$ and NADH in vitro. (**A**) Structures of ATP, NAD$^+$, and NADH. Grey, identical atoms; black, distinct atoms. (**B**) Processing of RNA 5' ends by RppH and NudC. A, adenosine; N$^+$, NAD$^+$ nicotinamide; N, NADH nicotinamide; p, phosphate. (**C** and **D**) NCIN capping with NAD$^+$ and NADH by *S. cerevisiae* mtRNAP (**C**) and human mtRNAP (**D**). Top, promoter derivatives. Middle, initial RNA products of in vitro transcription reactions with ATP, NAD$^+$, or NADH as initiating nucleotide and [α$^{32}$P]-GTP as extending nucleotide. Bottom, full-length RNA products of in vitro transcription reactions with ATP, NAD$^+$, or NADH as initiating nucleotide and [α$^{32}$P]-GTP, ATP, UTP, and 3'-deoxy-CTP (**C**), or [α$^{32}$P]-GTP, ATP, and UTP (**D**) as extending nucleotides. Products were treated with RppH or NudC as indicated. Grey box and arrow, transcription start site (TSS);+31 and+17/18, position of last NTP incorporated into full-length RNA products; M, 10-nt marker.

DOI: https://doi.org/10.7554/eLife.42179.002

The following source data and figure supplement are available for figure 1:

**Source data 1.** Source data for *Figure 1C,D*.
DOI: https://doi.org/10.7554/eLife.42179.004

**Source data 2.** Data for *Figure 1—figure supplement 1*.
DOI: https://doi.org/10.7554/eLife.42179.005

**Figure supplement 1.** *S. cerevisiae* and human mtRNAPs cap RNA with NAD$^+$ in vitro: additional data.
DOI: https://doi.org/10.7554/eLife.42179.003

performed reactions using either ATP, NAD$^+$, or NADH as the initiating entity and using [α$^{32}$P]-GTP as the extending nucleotide (*Figure 1C–D*, middle). We observed efficient formation of an initial RNA product in all cases (*Figure 1C–D*, middle). The initial RNA products obtained with ATP, but not with NAD$^+$ or NADH, were processed by RppH, which previous work has shown to process 5'-triphosphate RNAs to 5'-monophosphate RNAs (*Deana et al., 2008*) (*Figure 1B*), whereas the initial RNA products obtained with NAD$^+$ or NADH, but not with ATP, were processed by NudC, which previous work has shown to process 5'-NAD$^+$- and 5'-NADH-capped RNAs to 5'-monophosphate RNAs (*Cahová et al., 2015*; *Höfer et al., 2016*) (*Figure 1B*). The results establish that *S. cerevisiae* mtRNAP and human mtRNAP are able to generate initial RNA products using NAD$^+$ and NADH as NCINs.

We next assessed whether the initial RNA products formed using NAD$^+$ and NADH as NCINs can be extended to yield full-length RNA products (*Figure 1C–D*, bottom, and *Figure 1—figure supplement 1*). We performed parallel transcription experiments using either ATP, NAD$^+$, or NADH as the initiating entity and using [α$^{32}$P]-GTP, ATP, UTP, and 3'-deoxy-CTP (*Figure 1C*, bottom) or [α$^{32}$P]-GTP, ATP, and UTP (*Figure 1D*, bottom) as extending nucleotides. We observed efficient formation of full-length RNA products in all cases, and we observed that full-length RNA products obtained with NAD$^+$ or NADH, but not with ATP, were sensitive to NudC treatment (*Figure 1C–D*, bottom). We also observed efficient formation of full-length RNA products in transcription experiments performed using [α$^{32}$P]-ATP or [$^{32}$P]-NAD$^+$ as the initiating entity and using non-radiolabeled extending nucleotides (*Figure 1—figure supplement 1*). Full-length products obtained with [$^{32}$P]-NAD$^+$, but not with [α$^{32}$P]-ATP, were insensitive to treatment with alkaline phosphatase (which processes 5' phosphates). Furthermore, NudC treatment of full-length products obtained with [$^{32}$P]-NAD$^+$ yielded products that were sensitive to alkaline phosphatase (*Figure 1—figure supplement 1*). The results establish that *S. cerevisiae* mtRNAP and human mtRNAP not only generate initial RNA products, but also generate full-length RNA products, using NAD$^+$ and NADH as NCINs.

## *S. cerevisiae* and human mtRNAPs cap RNA with NAD$^+$ and NADH more efficiently than bacterial and nuclear RNAPs

We next determined the relative efficiencies of NCIN-mediated initiation vs. ATP-mediated initiation, $(k_{cat}/K_M)_{NCIN} / (k_{cat}/K_M)_{ATP}$, for mtRNAPs (*Figure 2* and *Figure 2—figure supplement 1*; methods as in *Bird et al., 2017*). We performed reactions with *S. cerevisiae* mtRNAP and DNA templates carrying the *S. cerevisiae* mitochondrial 21S promoter or 15S promoter (*Figure 2A* and *Figure 2—figure supplement 1*), and, in parallel, with human mtRNAP and DNA templates carrying the human mitochondrial light-strand promoter (LSP) or heavy-strand promoter (HSP1) (*Figure 2B* and *Figure 2—figure supplement 1*). We obtained values of $(k_{cat}/K_M)_{NCIN} / (k_{cat}/K_M)_{ATP}$ of ~0.3 to ~0.4 for NCIN-mediated initiation with NAD$^+$ and NADH by *S. cerevisiae* mtRNAP and ~0.2 to ~0.6 for NCIN-mediated initiation with NAD$^+$ and NADH by human mtRNAP. These values imply that NCIN-mediated initiation with NAD$^+$ or NADH is up to 40% as efficient as initiation with ATP for *S. cerevisiae* mtRNAP and up to 60% as efficient as initiation with ATP for human mtRNAP.

The observed efficiencies of NCIN-mediated initiation with NAD$^+$ or NADH by mtRNAPs are substantially higher than the highest previously reported efficiencies for NCIN-mediated initiation with NAD$^+$ or NADH by cellular RNAPs (~15%; *Bird et al., 2017*; *Bird et al., 2016*; *Vvedenskaya et al., 2018*). To enable direct comparison of efficiencies of NCIN capping by mtRNAPs vs. cellular RNAPs on the same templates under identical reaction conditions, we performed transcription assays using a 'tailed' template (*Figure 2C*, top) that bypasses the requirement for sequence-specific RNAP-DNA interactions and transcription-initiation factor-DNA interactions for transcription initiation (*Dedrick and Chamberlin, 1985*; *Kadesch and Chamberlin, 1982*). In these experiments, we observe efficiencies of NCIN-mediated initiation with NAD$^+$ and NADH by mtRNAP that are fully ~10 to ~40 fold higher than efficiencies of NCIN-mediated initiation with NAD$^+$ and NADH by *E. coli* RNAP and *S. cerevisiae* RNAP II (*Figure 2C*, bottom). We conclude that *S. cerevisiae* mtRNAP and human mtRNAP cap RNA with NAD$^+$ and NADH more efficiently than bacterial RNAP and eukaryotic nuclear RNAP II.

We next used the same tailed template and reaction conditions as in assays performed with mtRNAPs to determine the efficiency of NCIN-mediated initiation with NAD$^+$ and NADH for the single-subunit RNAP of bacteriophage T7 (T7 RNAP) (*Figure 2—figure supplement 2*). The efficiencies of NCIN-mediated initiation with NAD$^+$ and NADH by T7 RNAP were nearly as high as the

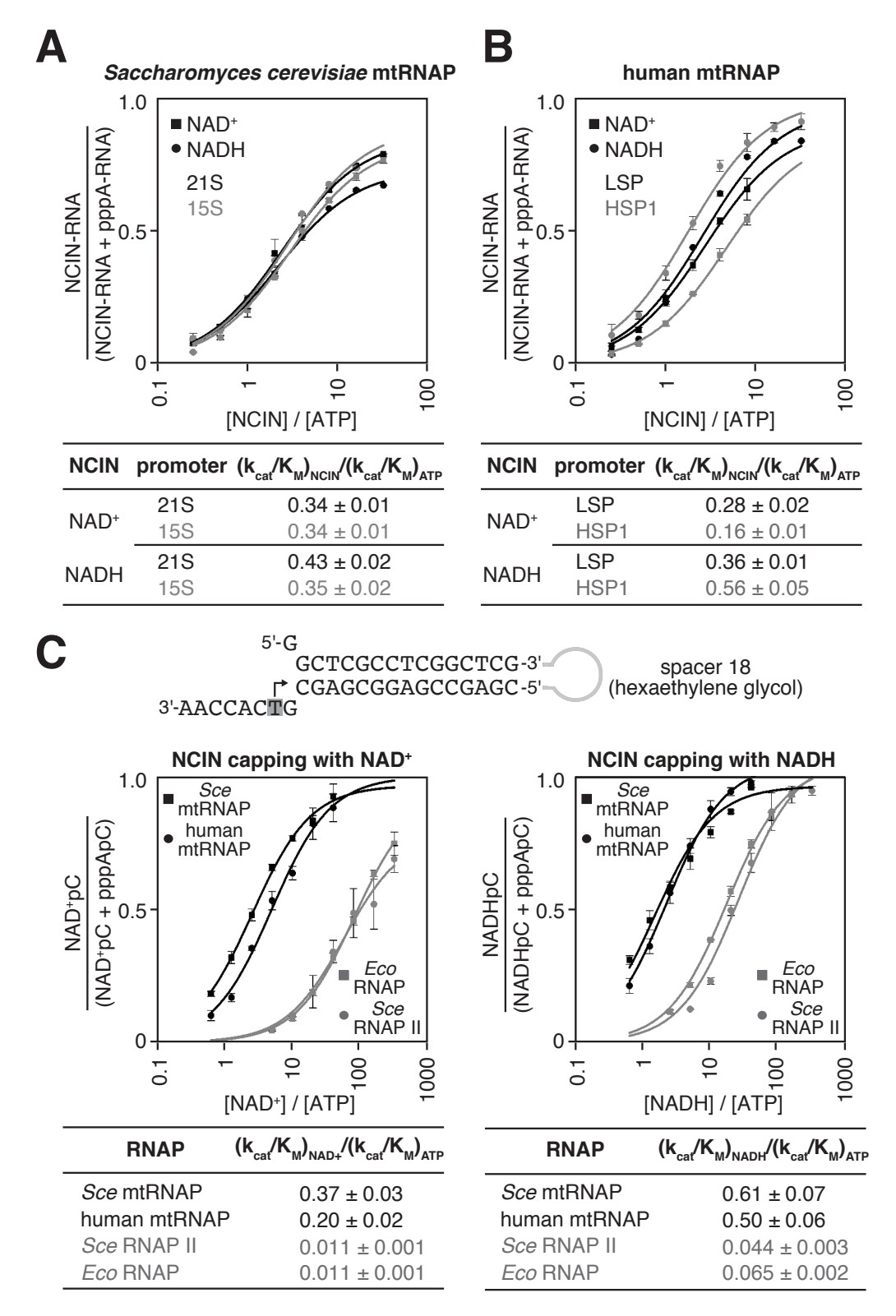

**Figure 2.** *S. cerevisiae* and human mtRNAPs cap RNA with NAD⁺ and NADH more efficiently than bacterial and nuclear RNAPs. (**A** and **B**) Dependence of NCIN-mediated capping with NAD⁺ and NADH on [NCIN] / [ATP] ratio for *S. cerevisiae* mtRNAP (**A**) and human mtRNAP (**B**) (mean ± SD; n = 3). DNA templates and representative data are shown in *Figure 2—figure supplement 1*. (**C**) Dependence of NCIN-mediated capping with NAD⁺ and NADH on [NCIN] / [ATP] ratio for mtRNAPs vs. *E. coli* RNAP and *S. cerevisiae* RNAP II. Top, tailed template. Grey box and arrow indicate TSS. Bottom, *Figure 2 continued on next page*

*Figure 2 continued*

dependence of NCIN-mediated capping with NAD$^+$ and NADH on [NCIN] / [ATP] ratio for *S. cerevisiae* mtRNAP (*Sce* mtRNAP), human mtRNAP, *E. coli* RNAP (*Eco* RNAP) and *S. cerevisiae* RNAP II (*Sce* RNAP II) (mean ± SD; n = 3).

DOI: https://doi.org/10.7554/eLife.42179.006

The following source data and figure supplements are available for figure 2:

**Source data 1.** Data for *Figure 2A,B*.
DOI: https://doi.org/10.7554/eLife.42179.009
**Source data 2.** Data for *Figure 2C,D*, and *Figure 2—figure supplement 1*.
DOI: https://doi.org/10.7554/eLife.42179.010
**Figure supplement 1.** Dependence of NCIN-mediated capping with NAD$^+$ and NADH on [NCIN] / [ATP] ratio for mtRNAPs: representative data.
DOI: https://doi.org/10.7554/eLife.42179.007
**Figure supplement 2.** *S. cerevisiae* and human mtRNAPs cap RNA with NAD$^+$ and NADH at least as efficiently as bacteriophage T7 RNAP.
DOI: https://doi.org/10.7554/eLife.42179.008

efficiencies of NCIN-mediated initiation by mtRNAPs. We conclude that there is a quantitative difference in the efficiency of NCIN capping between members of the single-subunit RNAP family (T7 RNAP and mtRNAPs) and members of the multi-subunit RNAP family (bacterial RNAP and eukaryotic nuclear RNAP II).

## Promoter sequence determines efficiency of RNA capping by mtRNAP

In previous work, we have shown that NCIN capping with NAD$^+$ and NADH by bacterial RNAP is determined by promoter sequence, particularly at and immediately upstream of, the transcription start site (TSS) (*Bird et al., 2016*; *Vvedenskaya et al., 2018*). NCIN capping by bacterial RNAP occurs only at promoters where the base pair (nontemplate-strand base:template-strand base) at the TSS is A:T (+1A promoters), and occurs most efficiently at the subset of +1A promoters where the base pair immediately upstream of the TSS is purine:pyrimidine (−1R promoters). We have further shown that sequence determinants for NCIN capping by bacterial RNAP reside within the template strand of promoter DNA (i.e., the strand that templates incoming nucleotide substrates) (*Vvedenskaya et al., 2018*).

To determine whether the specificity for A:T at the TSS (position +1), observed with bacterial RNAP, also is observed with mtRNAP, we assessed NAD$^+$ capping by *S. cerevisiae* mtRNAP using promoter derivatives having A:T or G:C at position +1 (*Figure 3A–B*). We observed NAD$^+$ capping in reactions performed using the promoter derivative having A:T at position +1, but not in reactions performed using the promoter derivative having G:C at position +1 (*Figure 3B*), indicating specificity for A:T at position +1. To determine whether specificity resides in the template strand for A:T at position +1, we analyzed NAD$^+$ capping with *S. cerevisiae* mtRNAP using template derivatives having noncomplementary nontemplate- and template-strand-nucleotides (A/C or G/T) at position +1 (*Figure 3B*). We observed NAD$^+$ capping only with the promoter derivative having T as the template strand base at position +1, indicating that specificity at position +1 resides in the template strand.

To determine whether specificity for R:Y at position −1, observed with bacterial RNAP, also is observed with mtRNAP, we analyzed NAD$^+$ capping by *S. cerevisiae* mtRNAP using promoter derivatives having either R:Y (A:T or G:C) or Y:R (C:G or T:A) at position −1 (*Figure 3C*). We observed higher efficiencies of NAD$^+$ capping with promoter derivatives having R:Y at position −1 than with promoter derivatives having Y:R (*Figure 3C*). To determine whether specificity at position −1 resides in the DNA template strand, we performed experiments using promoter derivatives having Y (C or T) or R (A or G) at position −1 of the template strand and having an abasic site (*) on the nontemplate strand (*Figure 3D*). We observed higher efficiencies of NAD$^+$ capping in reactions performed using promoter derivatives having Y at template-strand position −1 than with those having R. Furthermore, within error, the capping efficiencies for promoter derivatives having Y or R at template-strand position −1 matched the capping efficiencies for homoduplex promoter derivatives (*Figure 3C–D*), indicating that sequence information for NAD$^+$ capping with *S. cerevisiae* mtRNAP resides exclusively in the template strand.

We conclude that NCIN capping with NAD$^+$ by mtRNAP is determined by the sequence at, and immediately upstream of, the TSS (positions +1 and −1, respectively). We further conclude that the sequence and strand preferences at positions +1 and −1 for NCIN capping with NAD$^+$ by mtRNAP

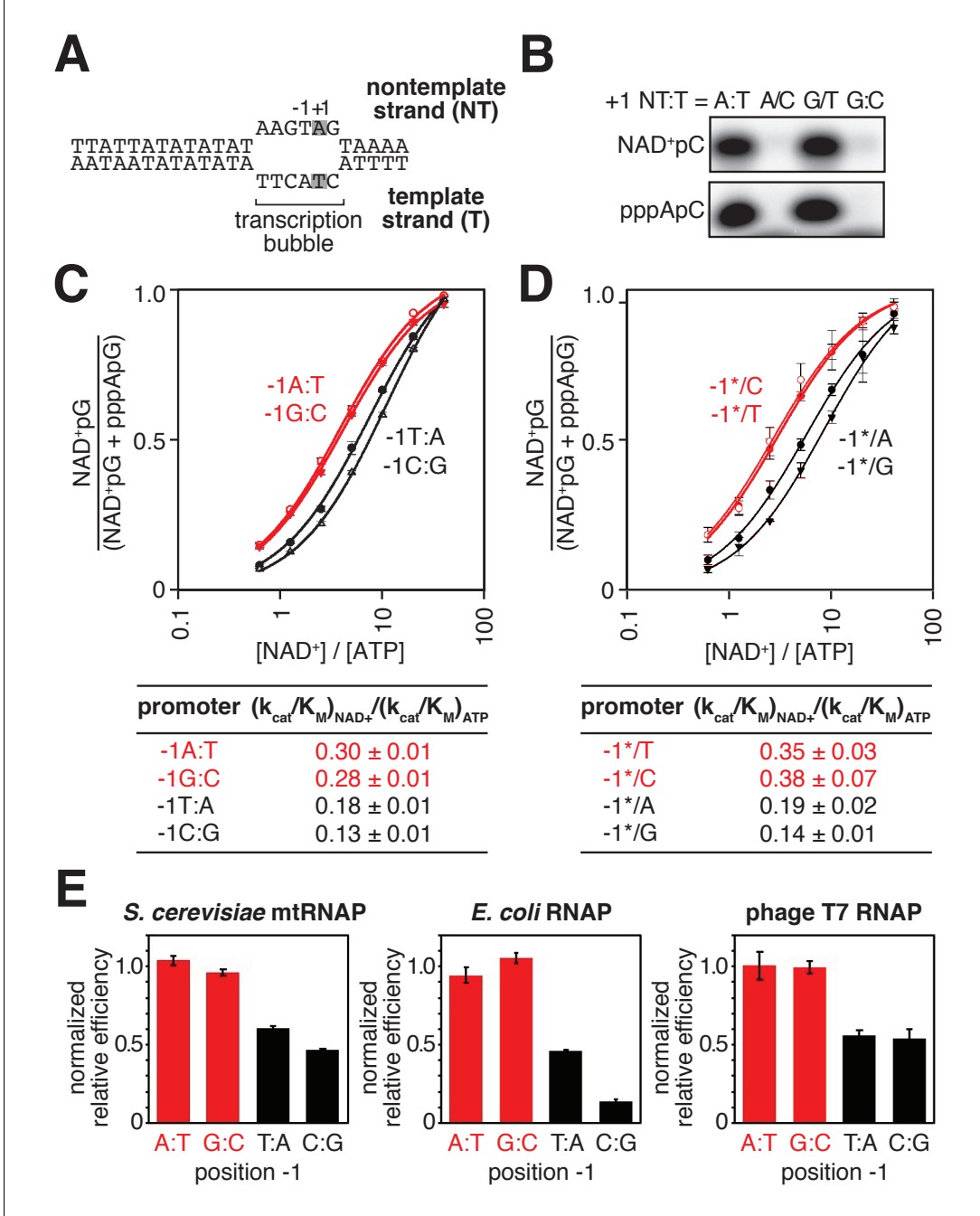

**Figure 3.** Promoter sequence determines efficiency of RNA capping with NAD⁺ by mtRNAP. (**A**) *S. cerevisiae* mitochondrial 21S promoter DNA depicted in the context of the mtRNAP-promoter open complex. DNA nontemplate strand (NT) on top; DNA template strand (T) on bottom; Unwound, non-base-paired DNA region, 'transcription bubble,' indicated by raised and lowered nucleotides; +1 and grey boxes, bases at the TSS; −1, bases immediately upstream of the TSS (the 21S promoter is a −1Y promoter). (**B**) Products of transcription reactions with NAD⁺ as initiating nucleotide and [α³²P]-CTP as extending nucleotide for templates having complementary or non-complementary nucleotides at position +1. (**C**) Dependence of NAD⁺ capping on [NAD⁺] / [ATP] ratio for homoduplex templates having A:T, G:C, T:A, or C:G at position −1 relative to TSS (mean ± SD; n = 3). Red, −1R promoters; black, −1Y promoters. (**D**) Dependence of NAD⁺ capping on [ATP] / [NAD⁺] ratio for heteroduplex templates having an abasic site (*) on the DNA nontemplate strand (mean ± SD; n = 3). Red, promoters with a template-strand Y; black, promoters with a template-strand R. (**E**) Sequence preferences at position −1 for *S. cerevisiae* mtRNAP, *E. coli* RNAP, and T7 RNAP. Graphs show normalized values of (k$_{cat}$/K$_M$)$_{NAD+}$ / (k$_{cat}$/K$_M$)$_{ATP}$ determined for homoduplex templates having A:T, G:C, T:A, or C:G at position −1 (mean ± SD; n = 3). Normalized values were calculated by dividing the value for each individual promoter by the average value measured for −1R promoters. Data for *S. cerevisiae* mtRNAP are from panel C, data for *E. coli* RNAP are from (**Vvedenskaya et al., 2018**), and data for T7 RNAP are from **Figure 3—figure supplement 1**.

*Figure 3 continued on next page*

*Figure 3 continued*

DOI: https://doi.org/10.7554/eLife.42179.011

The following source data and figure supplement are available for figure 3:

**Source data 1.** Data for *Figure 3B*.
DOI: https://doi.org/10.7554/eLife.42179.013
**Source data 2.** Data for *Figure 3C,D,E*, and *Figure 3—figure supplement 1*.
DOI: https://doi.org/10.7554/eLife.42179.014
**Figure supplement 1.** Promoter sequence determines efficiency of RNA capping with NAD$^+$: bacteriophage T7 RNAP.
DOI: https://doi.org/10.7554/eLife.42179.012

match the sequence and strand preferences observed for bacterial RNAP (*Figure 3C–E*) (*Bird et al., 2016*; *Vvedenskaya et al., 2018*), suggesting that these sequence and strand preferences may be universal determinants of NCIN capping with NAD$^+$ for all RNAPs. Consistent with this hypothesis, we find that sequence preferences for NCIN capping with NAD$^+$ by bacteriophage T7 RNAP, another member of the single-subunit RNAP family, match the sequence preferences observed for *S. cerevisiae* mtRNAP and bacterial RNAP (*Figure 3E* and *Figure 3—figure supplement 1*). Further consistent with this hypothesis, structural modeling suggests the basis for these sequence and strand preferences is universal: specifically, a strict requirement for template-strand +1T for base pairing to the NAD$^+$ adenine moiety, and a preference for template strand −1Y for 'pseudo' base pairing to the NAD$^+$ nicotinamide moiety (*Bird et al., 2016*; *Vvedenskaya et al., 2018*).

## Detection and quantitation of NAD$^+$- and NADH-capped mitochondrial RNA in vivo: boronate affinity electrophoresis with processed RNA and synthetic standards

Kössel, Jäschke and co-workers have demonstrated that boronate affinity electrophoresis allows resolution of uncapped RNAs from capped RNAs–such as m$^7$G, NAD$^+$ and NADH–that contain a vicinal-diol moiety (*Igloi and Kössel, 1985*; *Igloi and Kössel, 1987*; *Nübel et al., 2017*). However, the procedures of Kössel, Jäschke and co-workers have two limitations: (i) boronate affinity electrophoresis does not allow resolution of RNAs longer than ~200 nt, and (ii) boronate affinity electrophoresis, by itself, is unable to distinguish between different vicinal-diol containing cap structures (m$^7$G, NAD$^+$, NADH, or others). Here, to overcome these limitations, we have combined boronate affinity electrophoresis with use of oligodeoxynucleotide-mediated RNA cleavage ('DNAzyme' cleavage) (*Joyce, 2001*) and use of synthetic NCIN-capped RNA standards generated using NCIN-mediated transcription initiation in vitro (*Figure 4*). Use of DNAzyme cleavage enables processing of long RNAs to yield defined, short, 5'-end-containing subfragments (*Figure 4A*). Use of synthetic NCIN-capped RNA standards enables distinction between capped species (*Figure 4A,D*).

To detect and quantify NCIN capping with NAD$^+$ and NADH in mitochondrial RNA isolated from cells, we employed the following steps: (i) DNAzyme cleavage of target RNAs to generate 5'-end-containing subfragments < 80 nt in length (*Figure 4A*, top); (ii) DNAzyme treatment of synthetic NAD$^+$- and NADH-capped RNA standards having sequences identical to RNAs of interest (*Figure 4A*, bottom); (iii) boronate affinity electrophoresis of DNAzyme-generated subfragments of mitochondrial RNA and DNAzyme-generated subfragments of synthetic NAD$^+$- and NADH-capped RNA standards having sequences identical to RNAs of interest (*Figure 4B*); and (iv) detection of DNAzyme-generated 5'-end-containing subfragments of mitochondrial RNAs and synthetic RNA standards by hybridization with a radiolabeled oligodeoxyribonucleotide probe (*Figure 4C–D*).

We selected for analysis two *S. cerevisiae* mitochondrial RNAs that previously had been detected as NAD$^+$-capped: COX2 and 21S (*Walters et al., 2017*). We isolated *S. cerevisiae* total RNA and analyzed COX2 and 21S RNAs using the procedure described in the preceding paragraph. The results are presented in *Figures 4C–D* and *5A* (top). For both COX2 and 21S RNAs, we detect at least one RNA species with electrophoretic mobility retarded as compared to that of uncapped RNA, indicating the presence of capped RNA. Treatment with the decapping enzyme NudC eliminates these species, confirming the presence of capping. Comparison of the electrophoretic mobilities to those of synthetic NAD$^+$- and NADH-capped RNA standards indicates that one of the capped species is NAD$^+$-capped RNA, and the other capped species, present under these growth

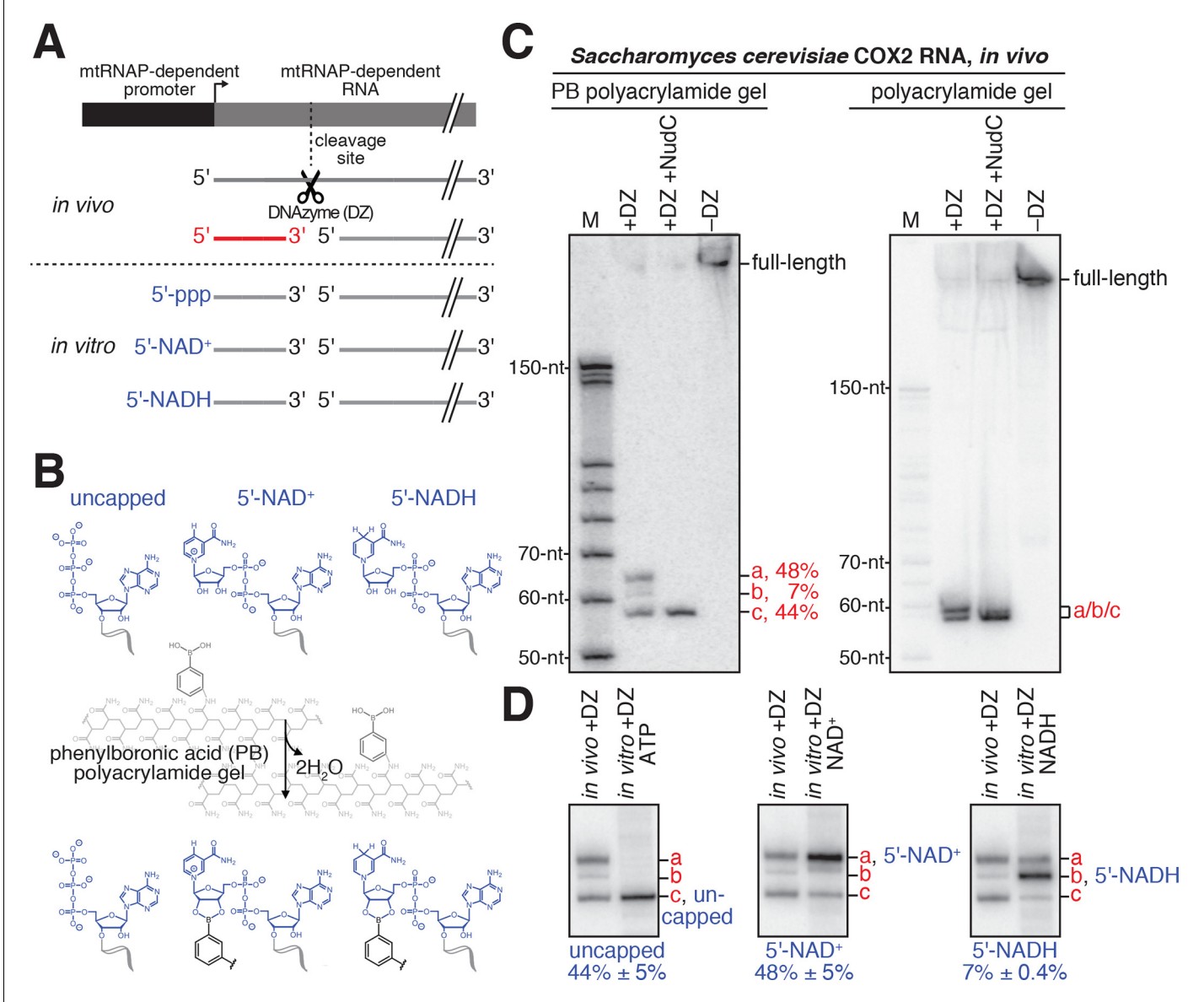

**Figure 4.** Detection and quantitation of NAD⁺- and NADH-capped mitochondrial RNA in vivo: boronate affinity electrophoresis with DNAzyme-cleaved cellular RNA and DNAzyme-cleaved synthetic NCIN-capped RNA standards. (**A**) Use of DNAzyme (DZ) to process RNA to yield a defined, short 5'-end-containing subfragment, in parallel in vivo (red) and in vitro (blue). Uncapped, 5'-triphosphate (ppp) end generated using ATP-mediated initiation; 5'-NAD⁺, NAD⁺-capped end generated using NAD⁺-mediated initiation; 5'-NADH, NADH-capped end generated using NADH-mediated initiation. (**B**) Use of boronate affinity electrophoresis to resolve 5'-uncapped, 5'-NAD⁺, and 5'-NADH containing RNAs. Grey, structure of phenylboronic acid (PB) polyacrylamide gel. (**C**) PB-polyacrylamide gel (left) and polyacrylamide gel (right) analysis of DNAzyme-generated 5'-end-containing subfragments of *S. cerevisiae* mitochondrial RNA COX2. Red, observed 5'-end-containing RNA subfragments resolved by PB-polyacrylamide-gel (left) or not resolved by polyacrylamide gel (right); identities of these subfragments are defined in Panel D. (**D**) Comparison of electrophoretic mobilities of observed 5'-end-containing subfragments of COX2 RNA generated in vivo to 5'-end-containing subfragments of synthetic RNA standards generated in vitro. a, NAD⁺-capped RNA; b, NADH-capped RNA; c, uncapped RNA (mean ± SD; n = 3).
DOI: https://doi.org/10.7554/eLife.42179.015

The following source data is available for figure 4:

**Source data 1.** Data for *Figure 4D*.
DOI: https://doi.org/10.7554/eLife.42179.016

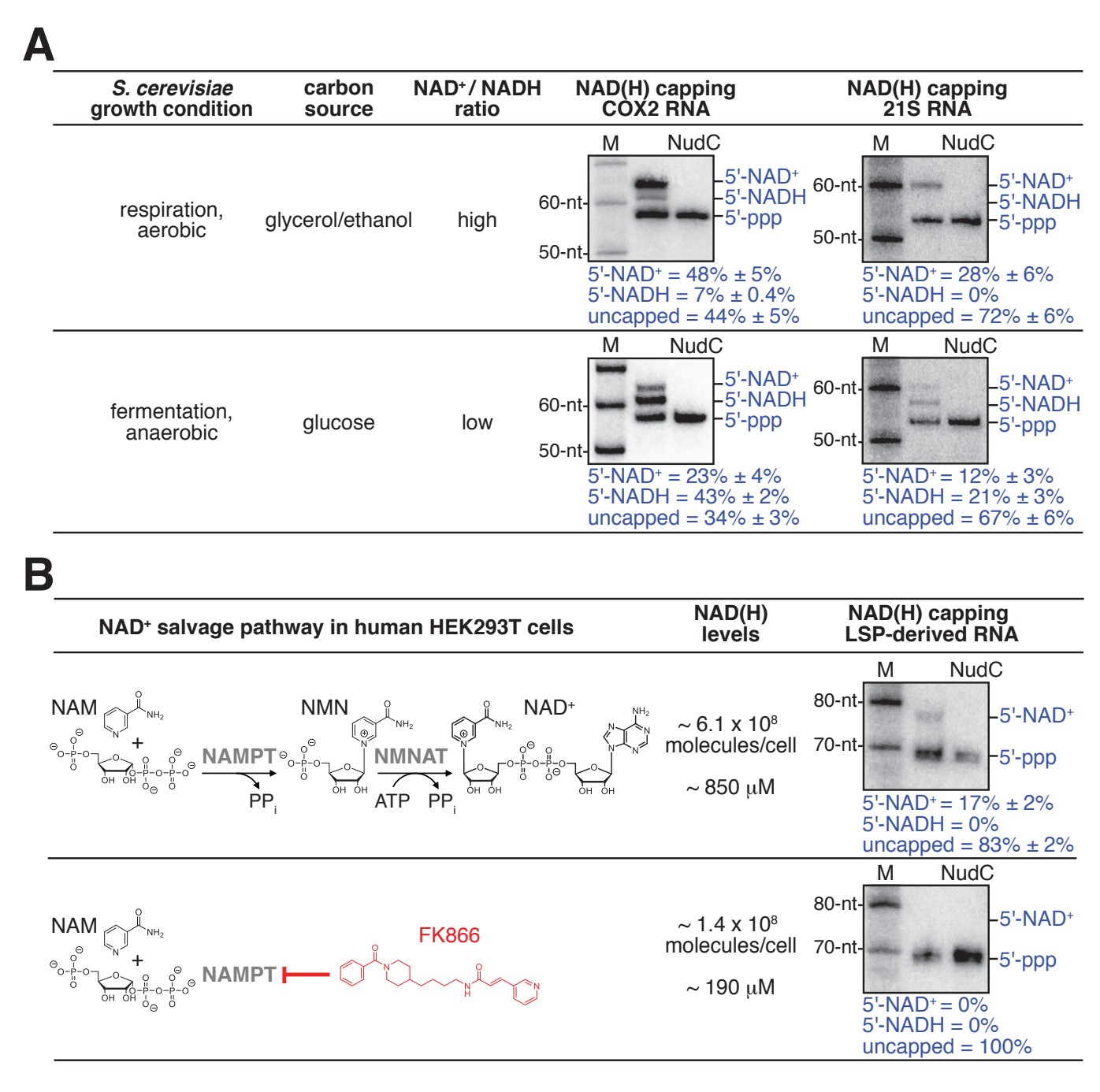

**Figure 5.** Detection and quantitation of NAD[+]- and NADH-capped mitochondrial RNA in vivo: effects of intracellular NAD[+] and NADH levels in *S. cerevisiae* and human cells. (**A**) Changes in intracellular NAD[+]/NADH ratios result in changes in levels of NAD[+]- and NADH-capped mitochondrial RNA (mean ± SD; n = 3). Gel images show representative data for *S. cerevisiae* COX2 RNA (left) and 21S RNA (right). Blue annotations as in *Figure 4*. (**B**) Changes in intracellular NAD(H) levels result in changes in levels of NAD[+]- and NADH-capped mitochondrial RNA (mean ± SD; n = 3). Gel images show representative data for LSP-derived RNAs. Red, NAD(H) biosynthesis inhibitor FK866; NAMPT, Nicotinamide phosphoribosyltransferase; NMNAT, Nicotinamide mononucleotide adenylyltransferase.

DOI: https://doi.org/10.7554/eLife.42179.017

The following source data is available for figure 5:

**Source data 1.** Data for *Figure 5* (gels).
DOI: https://doi.org/10.7554/eLife.42179.018
**Source data 2.** Data for *Figure 5* (values of NCIN capping).
DOI: https://doi.org/10.7554/eLife.42179.019

conditions only for COX2 RNA, is NADH-capped RNA. The results show that both COX2 and 21S RNAs are present in NAD$^+$-capped forms, and that COX2 RNA also is present in an NADH-capped-form. Because the hybridization probe detects RNA fragments that contain 5' ends generated by transcription initiation (red subfragment depicted in *Figure 4A*), the detected NAD$^+$ and NADH caps are concluded to be at 5' ends generated by transcription initiation, as opposed to 5' ends generated by RNA processing. The results confirm that *S. cerevisiae* mitochondrial RNAs undergo NAD$^+$ capping in cells, show that *S. cerevisiae* mitochondrial RNAs undergo NAD$^+$ capping at 5' ends generated by transcription initiation (as opposed to 5' ends generated by RNA processing), and show that *S. cerevisiae* mitochondrial RNAs also undergo NADH capping in cells.

The observed levels of NAD$^+$- and NADH-capping of *S. cerevisiae* mitochondrial RNAs are remarkably high. For COX2 RNA, NAD$^+$-capped RNA comprises ~50% of the total COX2 RNA pool and NADH-capped RNA comprises ~10% of the total COX2 RNA pool (*Figures 4C–D* and *5A*, top). For 21S RNA, NAD$^+$-capped RNA comprises ~30% of the total 21S RNA pool (*Figure 5A*, top). These levels of NCIN capping are ~2 to ~50 times higher than levels of NCIN-capping in exponentially growing *E. coli* (less than 1% to ~20% for NAD$^+$-capping; not previously detected for NADH-capping) (*Bird et al., 2016*; *Cahová et al., 2015*; *Nübel et al., 2017*; *Vvedenskaya et al., 2018*).

We performed analogous experiments analyzing RNAs produced by transcription from the human mitochondrial LSP promoter (*Figure 5B*, top). We isolated and analyzed total RNA from HEK293T cells. We observed an NAD$^+$-capped species comprising ~15% of the total LSP-derived RNA pool (*Figure 5B*, top). The results establish that human mitochondrial RNAs undergo NAD$^+$ capping in cells and show that human mitochondrial RNAs undergo NAD$^+$ capping at 5' ends generated by transcription initiation (as opposed to 5' ends generated by RNA processing).

## Detection and quantitation of NAD$^+$- and NADH-capped mitochondrial RNA in vivo: mtRNAPs serve as both sensors and actuators in coupling cellular metabolism to mitochondrial gene expression

Mitochondria are the primary locus of metabolism and energy transformation in the eukaryotic cell, serving as the venue for the tricarboxylic acid cycle (TCA) cycle and oxidative phosphorylation. The TCA cycle reduces NAD$^+$ to NADH and oxidative phosphorylation oxidizes NADH to NAD$^+$. Our finding that mtRNAPs perform NCIN capping with NAD$^+$ and NADH at efficiencies that vary in a simple mass-action-dependent fashion with [NAD$^+$] / [ATP] and [NADH] / [ATP] ratios in vitro (*Figures 1* and *2*), and our finding that mtRNAPs perform efficient NCIN capping to yield NAD$^+$- and NADH-capped mitochondrial RNAs in vivo (*Figures 4* and *5*), raise the possibility that mtRNAPs may serve as both sensors and actuators in coupling metabolism to mitochondrial gene expression in vivo.

As a first test of this hypothesis, we assessed whether changing intracellular [NAD$^+$] / [NADH] ratios results in changes in NAD$^+$ and NADH capping of mitochondrial RNAs. We isolated total RNA from *S. cerevisiae* grown under conditions that result in either high or low [NAD$^+$] / [NADH] ratios (*Bekers et al., 2015*; *Canelas et al., 2008*): respiration (glycerol/ethanol; aerobic) or fermentation (glucose; anaerobic). We analyzed the same two mitochondrial RNAs as above: COX2 and 21S (*Figure 5A*). We observed marked changes in levels of NAD$^+$ and NADH capping for both analyzed mitochondrial RNAs. For COX2 RNA, on changing from the growth condition yielding a high [NAD$^+$] / [NADH] ratio to the growth condition yielding a low [NAD$^+$] / [NADH] ratio, we observe a decrease in levels of NAD$^+$ capping (from ~50% to ~20%) and an anti-correlated increase in the level of NADH capping (from ~10% to ~40%). Notably, the total level of NAD$^+$ and NADH capping, NAD(H) capping, remains constant under the two conditions (~60%), indicating that the relative levels of NCIN-mediated initiation and ATP-mediated initiation do not change (*Figure 5A*). For 21S RNA, the same pattern is observed (*Figure 5A*): on changing from the growth condition yielding a high [NAD$^+$] / [NADH] ratio to the growth condition yielding a low [NAD$^+$] / [NADH] ratio, the level of NAD$^+$ capping decreases (from ~30% to ~10%), the level of NADH capping increases (from 0% to ~20%), and the total level of NAD$^+$ and NADH capping remains constant (~30%). The results indicate that changing the [NAD$^+$] / [NADH] ratio changes transcription outputs in vivo.

As a second test of this hypothesis, we assessed whether changing intracellular total NAD(H) levels results in changes in NCIN capping of mitochondrial RNAs (*Figure 5B*). We isolated RNA from human HEK293T cells grown under conditions yielding either high or low intracellular NAD(H) levels: standard growth media or growth media in the presence of the NAD(H)-biosynthesis inhibitor FK866 (*Hasmann and Schemainda, 2003*; *Khan et al., 2006*). We analyzed the same LSP-derived mitochondrial RNA as in the preceding section (*Figure 5B*). On changing from the growth condition yielding high intracellular NAD(H) levels to the growth condition yielding low NAD(H) levels, we observe a marked change in the total level of NCIN capping (from ~15% to 0%). The results indicate that changing NAD(H) levels changes levels of NCIN-capped mitochondrial RNAs in vivo.

Taken together, the results of the two experiments in *Figure 5* indicate that mtRNAP serves as both sensor and actuator in coupling [NAD$^+$] / [NADH] ratios to relative levels of NAD$^+$- and NADH-capped mitochondrial RNAs (*Figure 5A*), and in coupling total NAD(H) levels to total levels of NCIN-capped mitochondrial RNAs (*Figure 5B*), thereby coupling cellular metabolism to mitochondrial transcription outputs. We suggest that mtRNAPs serve as sensors through their mass-action-dependence in selecting NAD$^+$ vs. NADH vs. ATP as the initiating entity during transcription initiation, and serve as actuators by incorporating NAD$^+$ vs. NADH vs. ATP at the RNA 5' end during transcription initiation.

## Discussion

Our results show that *S. cerevisiae* and human mtRNAPs cap RNA with NAD$^+$ and NADH (*Figure 1*), show that *S. cerevisiae* and human mtRNAPs cap RNA with NAD$^+$ and NADH more efficiently than bacterial and eukaryotic nuclear RNAPs (*Figure 2*), and show that capping efficiency by mtRNAPs is determined by promoter sequence (*Figure 3*). Our results further show that the proportions of mitochondrial RNAs that are capped with NAD$^+$ and NADH are remarkably high–up to ~60% (*Figures 4* and *5*)–and that these proportions change in response to cellular NAD$^+$ and NADH levels (*Figure 5*).

We and others previously have shown that NCIN capping by cellular RNAPs has functional consequences, including modulating RNA stability and modulating RNA translatability (*Bird et al., 2016*; *Cahová et al., 2015*; *Jiao et al., 2017*). Our results here showing that *S. cerevisiae* and human mitochondrial RNAs are capped at substantially higher levels than non-mitochondrial RNAs–up to ~60% for analyzed *S. cerevisiae* mitochondrial RNAs and up to ~15% for analyzed human mitochondrial RNAs (*Figures 4* and *5*)–suggest that NCIN capping in mitochondria occurs at a higher efficiency, and has a higher importance, than NCIN capping in other cellular compartments. Four other considerations support this hypothesis. First, mtRNAPs are substantially more efficient at NAD$^+$ and NADH capping than bacterial and eukaryotic nuclear RNAPs (*Figure 2C*). Second, levels of NAD$^+$ and NADH relative to ATP in mitochondria are substantially higher than levels of NAD$^+$ and NADH relative to ATP in bacteria and eukaryotic nuclei (*Cambronne et al., 2016*; *Chen et al., 2016*; *Park et al., 2016*). Third, all *S. cerevisiae* and human mitochondrial promoters are +1A promoters (18 promoters in *S. cerevisiae* mitochondria; two promoters in human mitochondria) (*Biswas, 1999*; *Chang and Clayton, 1984*; *Taanman, 1999*), in contrast to bacterial and eukaryotic nuclear RNAP promoters, for which approximately half are +1A promoters (*Chen et al., 2013*; *Haberle et al., 2014*; *Saito et al., 2013*; *Thomason et al., 2015*; *Tsuchihara et al., 2009*). Fourth, we observe capping with both NAD$^+$ and NADH for mitochondrial RNAs in vivo (*Figures 4* and *5*), whereas, to date, capping has been observed with only NAD$^+$ for non-mitochondrial RNAs in vivo, raising the possibility that, in mitochondria, but not in other cellular compartments, NAD$^+$ and NADH caps dictate different RNA fates and, correspondingly, different transcription outputs.

Our results showing that levels of NAD$^+$ and NADH capping by mtRNAP correlate with changes in intracellular levels of NAD$^+$ and NADH (*Figure 5*) indicate that mtRNAP serves as a sensor, sensing [NAD$^+$] / [ATP] and [NADH] / [ATP] ratios, when selecting initiating entities and, simultaneously, serves as an actuator by altering RNA 5' ends when selecting initiating entities. Because all *S. cerevisiae* and human mitochondrial promoters are +1A promoters, this dual sensor/actuator activity of mtRNAP will occur at, and couple metabolism to gene expression at, all *S. cerevisiae* and human mitochondrial promoters. This dual sensor/actuator activity of mtRNAP obviates the need for a dedicated signal-processing machinery for coupling metabolism to gene expression, instead employing a pan-eukaryotic housekeeping enzyme for signal processing. As such, this dual sensor/actuator

activity of mtRNAP provides a remarkably economical, parsimonious mechanism of coupling metabolism to gene expression.

# Materials and methods

**Key resources table**

| Reagent type (species) or resource | Designation | Source or reference | Identifiers | Additional information |
|---|---|---|---|---|
| Strain, strain background (*E. coli*) | BL21(DE3) bacteria | NEB | C2527H | |
| Strain, strain background (*E. coli*) | NiCo21(DE3) bacteria | NEB | C2529H | |
| Strain, strain background (*E. coli*) | Artic Express (DE3) bacteria | Fisher Scientific | NC9444283 | |
| Strain, strain background (*S. cerevisiae*) | 246.1.1 (*MATa ura3 trp1 leu2 his4*) | Gift of A. Vershon | | |
| Cell line (human) | HEK293T (human embryonic kidney cells) | ATCC | CRL-3216 | |
| Recombinant DNA reagent | pIA900 | Gift of I. Artsimovitch | | |
| Recombinant DNA reagent | pET NudC-His | (*Bird et al., 2016*) | | |
| Recombinant DNA reagent | pJJ1399 | gift of J. Jaehning | | |
| Recombinant DNA reagent | pTrcHisC-Mtf1 | gift of J. Jaehning | | |
| Recombinant DNA reagent | pPROEXHTb-POLRMT (43–1230)−6xHis | (*Ramachandran et al., 2017*) | | |
| Recombinant DNA reagent | pPROEXHTb-TFAM (43-245)−6xHis | (*Ramachandran et al., 2017*) | | |
| Recombinant DNA reagent | pT7TEV-HMBP4 | (*Yakubovskaya et al., 2014*) | | |
| Recombinant DNA reagent | pAR1219 | (*Jia et al., 1996*) | | |
| Sequence-based reagent | DK64 | Integrated DNA Technologies (IDT) | tailed template with PEG$_6$ linker | GGCTCGCCTCGGCTCG/iSp18/CGAGCCGAGGCGAGCGTCACCAA |
| Sequence-based reagent | JB459 | IDT | human LSP DNA template + 1 AGU variant nontemplate strand | GTGTTAGTTGGGGGGGTGACTGTTAAAAGTGCATACCGCCAAAGTATAAAATTTGTGGGCC |
| Sequence-based reagent | JB460 | IDT | human LSP DNA template + 1 AGU variant template strand | GGCCCACAAATTTTATACTTTGGCGGTATGCACTTTTAACAGTCACCCCCCAACTAACAC |
| Sequence-based reagent | JB469 | IDT | T7φ2.5–35 n nontemplate strand (−1T) | CAGTAATACGACTCACTATTAGCGAAGCGGGCATGCGGCCAGCCATAGCCGATCA |
| Sequence-based reagent | JB470 | IDT | T7φ2.5–35 n template strand (−1A) | TGATCGGCTATGGCTGGCCGCATGCCCGCTTCGCTAATAGTGAGTCGTATTACTG |
| Sequence-based reagent | JB471 | IDT | T7φ2.5–35 n nontemplate strand (−1A) | CAGTAATACGACTCACTATAAGCGAAGCGGGCATGCGGCCAGCCATAGCCGATCA |

*Continued on next page*

*Continued*

| Reagent type (species) or resource | Designation | Source or reference | Identifiers | Additional information |
|---|---|---|---|---|
| Sequence-based reagent | JB472 | IDT | T7φ2.5–35 n template strand (−1T) | TGATCGGCTATGGCTGGCCGCATGCCC GCTTCGCTTATAGTGAGTCGTATTACTG |
| Sequence-based reagent | JB473 | IDT | T7φ2.5–35 n nontemplate strand (−1G) | CAGTAATACGACTCACTATGAGCGAAG CGGGCATGCGGCCAGCCATAG CCGATCA |
| Sequence based reagent | JB474 | IDT | T7φ2.5 35 n template strand (−1C) | TGATCGGCTATGGCTGGCCGCATGCC CGCTTCGCTCATAGTGAGTCGTATTACTG |
| Sequence-based reagent | JB475 | IDT | T7φ2.5–35 n nontemplate strand (−1C) | CAGTAATACGACTCACTATCAGCGAA GCGGGCATGCGGCCAGCCA TAGCCGATCA |
| Sequence-based reagent | JB476 | IDT | T7φ2.5–35 n template strand (−1G) | TGATCGGCTATGGCTGGCCGCATGC CCGCTTCGCTGATAGTG AGTCGTATTACTG |
| Sequence-based reagent | JB515 | IDT | probe for human LSP-generated RNA (complementary to positions + 2 to+31) | CACCAGCCTAACCAGATTTCAA ATTTTATC |
| Sequence-based reagent | JB525 | IDT | probe for *S. cerevisiae* 21S RNA (complementary to positions + 9 to+42) | CTATATAATAAATATTTCAAATC TATTATTCTAC |
| Sequence-based reagent | JB526 | IDT | *S. cerevisiae* 21S RNA DNAzyme; cleaves transcript at position + 53 | ACTCCATGATTAGGCTAGCTACAA CGACTCTTTAAATCT |
| Sequence-based reagent | JB555 | IDT | probe for *S. cerevisiae* COX2 RNA (complementary to positions + 8 to+46) | ATCTTAACCTTTAGACTCTTTTGTC TATTTATAATATGT |
| Sequence-based reagent | JB557 | IDT | *S. cerevisiae* COX2 DNAzyme; cleaves at position + 57 | TCTTAATAAATCTAAGGCTAGCTACA ACGAATTTTAATAAATCTT |
| Sequence-based reagent | JB559 | IDT | human LSP-generated RNA DNAzyme; cleaves at position + 67 | GCACTTAAACAGGCTAGCTACAA CGAATCTCTGCCA |
| Sequence-based reagent | JB560 | IDT | *S. cerevisiae* COX2 −40 to + 125 nontemplate strand oligo (for generation of in vitro transcription template) | TATATAATAATAAATTATAAATAAATTTT AATTAAAAGTAGTATTAACATATTATAAA TAGACAAAAGAGTCTAAAGGTTAAGATT TATTAAAATGTTAGATTTATTAAGATTAC AATTAACAAC |
| Sequence-based reagent | JB561 | IDT | *S. cerevisiae* COX2 −40 to + 3 forward primer (for generation of in vitro transcription template) | TATATAATAATAAATTATAAATAAATTTT AATTAAAAGTAGT |
| Sequence-based reagent | JB562 | IDT | *S. cerevisiae* COX2 + 83 to+125 reverse primer (for generation of in vitro transcription template) | GTTGTTAATTGTAATCTTAATAAATCTAA CATTTTAATAAATC |
| Sequence-based reagent | UB1 | IDT | human LSP DNA template (−43 to + 19) nontemplate strand | ATGTGTTAGTTGGGGGGTGACTGTTAA AAGTGCATACCGCCAAAAGATAAAATT TGAAATCTG |
| Sequence-based reagent | UB2 | IDT | human LSP DNA template (−43 to + 19) template strand | CAGATTTCAAATTTTATCTTTTGGCGGT ATGCACTTTTAACAGTCACCCCCCAAC TAACACAT |
| Sequence-based reagent | UB3 | IDT | human HSP1 DNA template (−43 to + 20) nontemplate strand | ACACACCGCTGCTAACCCCATACCCCGA ACCAACCAAACCCCAAAGACACCCGCC ACAGTTTA |
| Sequence-based reagent | UB4 | IDT | human HSP1 DNA template (−43 to + 20) template strand | TAAACTGTGGCGGGTGTCTTTGGGGT TTGGTTGGTTCGGGGTATGGGGTTA GCAGCGGTGTGT |

*Continued on next page*

*Continued*

| Reagent type (species) or resource | Designation | Source or reference | Identifiers | Additional information |
|---|---|---|---|---|
| Sequence-based reagent | UB5 | IDT | *S. cerevisiae* 15S DNA template (−25 to + 1; C-less cassette) nontemplate strand | ATAATTTATTTATTATTATATAAGTAATAAATAATTGTTTTATATAATAAGAATTCTCCTTC |
| Sequence-based reagent | UB6 | IDT | *S. cerevisiae* 15S DNA template (−25 to + 1; C-less cassette) template strand | GAAGGAGAATTCTTATTATATAAAACAATTATTTATTACTTATATAATAATAAATAAATTAT |
| Sequence-based reagent | UB7 | IDT | *S. cerevisiae* 21S DNA template (−25 to + 1; C-less cassette) nontemplate strand | TATTATTATTTATTATATATATAAGTAGTAAAAAGTAGAATAATAGATTTGAAATACC |
| Sequence-based reagent | UB8 | IDT | *S. cerevisiae* 21S DNA template (−25 to + 1; C-less cassette) template strand | GAAGGAGACCAACCACAAACACACAACAACCACCAACTACTTATATAATAATAAATAAATTAT |
| Sequence-based reagent | UB9 | IDT | *S. cerevisiae* 21S DNA template (−25 to + 1; C-less and A-less cassette) nontemplate strand | ATAATTTATTTATTATTATATAAGTAGTTGGTGGTTGTTGTGTGTTTGTGGTTGGTCTCCTTC |
| Sequence-based reagent | UB10 | IDT | *S. cerevisiae* 21S DNA template (−25to + 1; C-less and A-less cassette) template strand | GAAGGAGACCAACCACAAACACACAACAACCACCAACTACTTATATAATAATAAATAAATTAT |
| Sequence-based reagent | UB11 | IDT | *S. cerevisiae* 21S nontemplate strand (−1A) | ATAATTTATTTATTATTATATAAGAAGTTGGTGGTTGTTGTGTGTTTGTGGTTGGTCTCCTTC |
| Sequence-based reagent | UB12 | IDT | *S. cerevisiae* 21S template strand (−1T) | GAAGGAGACCAACCACAAACACACAACAACCACCAACTTCTTATATAATAAATAAATTAT |
| Sequence-based reagent | UB13 | IDT | *S. cerevisiae* 21S nontemplate strand (−1G) | ATAATTTATTTATTATTATATAAGGAGTTGGTGGTTGTTGTGTGTTTGTGGTTGGTCTCCTTC |
| Sequence-based reagent | UB14 | IDT | *S. cerevisiae* 21S template strand (−1C) | GAAGGAGACCAACCACAAACACACAACAACCACCAACTCCTTATATAATAATAAATAAATTAT |
| Sequence-based reagent | UB15 | IDT | *S. cerevisiae* 21S nontemplate strand (−1C) | ATAATTTATTTATTATTATATAAGCAGTTGGTGGTTGTTGTGTGTTTGTGGTTGGTCTCCTTC |
| Sequence-based reagent | UB16 | IDT | *S. cerevisiae* 21S template strand (−1G) | GAAGGAGACCAACCACAAACACACAACAACCACCAACTGCTTATATAATAATAAATAAATTAT |
| Sequence-based reagent | UB17 | IDT | *S. cerevisiae* 21S nontemplate strand (+1C) | ATAATTTATTTATTATTATATAAGTCGTTGGTGGTTGTTGTGTGTTTGTGGTTGGTCTCCTTC |
| Sequence-based reagent | UB18 | IDT | *S. cerevisiae* 21S template strand (+1G) | GAAGGAGACCAACCACAAACACACAACAACCACCAACGACTTATATAATAATAAATAAATTAT |
| Sequence-based reagent | JB527 | IDT | *S. cerevisiae* 21S nontemplate strand (−1 abasic) | ATAATTTATTTATTATTATATAAG/idSp/AGTTGGTGGTTGTTGTGTGTTTGTGGTTGGTCTCCTTC |
| Peptide, recombinant protein (*S. cerevisiae*) | Rpo41 (mtRNAP) | (*Tang et al., 2009*) | | |
| Peptide, recombinant protein (*S. cerevisiae*) | Mtf1 | (*Paratkar and Patel, 2010*) | | |
| Peptide, recombinant protein (Human) | POLRMT (mtRNAP) | (*Ramachandran et al., 2017*) | | |

*Continued on next page*

*Continued*

| Reagent type (species) or resource | Designation | Source or reference | Identifiers | Additional information |
|---|---|---|---|---|
| Peptide, recombinant protein (Human) | TFAM | (*Ramachandran et al., 2017*) | | |
| Peptide, recombinant protein (Human) | TFB2 | (*Yakubovskaya et al., 2014*) | | |
| Peptide, recombinant protein (*S. cerevisiae*) | RNA polymerase II | Gift of C. Kaplan | | |
| Peptide, recombinant protein (*E. coli*) | RNA polymerase core (β'−6xHis) | (*Artsimovitch et al., 2003*) | | |
| Peptide, recombinant protein | T7 RNA polymerase | (*Jia et al., 1996*) | | |
| Peptide, recombinant protein (*E. coli*) | NudC | (*Cahová et al., 2015*) | | |
| Peptide, recombinant protein | Phusion Flash HF master mix | ThermoFisher | F-548L | |
| Peptide, recombinant protein | T4 Polynucleotide Kinase | NEB | M0201L | |
| Peptide, recombinant protein | RNA 5' pyrophosphohydrolase (RppH) | NEB | M0356S | |
| Peptide, recombinant protein | FastAP Alkaline Phosphatase | Thermo Fisher | EF0651 | |
| Commercial assay or kit | Monarch PCR and DNA clean up kit | NEB | T1030S | |
| Chemical compound, drug | Nuclease-free water (not DEPC-treated) | ThermoFisher | AM9932 | |
| Chemical compound, drug | Bacto agar | VWR | 90000–760 | |
| Chemical compound, drug | Bacto tryptone | VWR | 90000–286 | |
| Chemical compound, drug | Bacto yeast extract | VWR | 90000–726 | |
| Chemical compound, drug | D-Glucose monhydrate | Amresco | 0643–1 kg | |
| Chemical compound, drug | Glycerol | EMD Millipore | 55069521 | |
| Chemical compound, drug | DMEM medium | Thermo Fisher | 11965–092 | |
| Chemical compound, drug | Fetal Bovine Serum | Atlanta Biological | S11150H | |
| Chemical compound, drug | dNTP solution mix, 10 mM of each NTP | NEB | N0447S | |
| Chemical compound, drug | NTP set (ultra-pure), 100 mM solutions | GE Healthcare | 27-2025-01 | |
| Chemical compound, drug | NAD$^+$ | Roche (Sigma-Aldrich) | 10127965001 | |
| Chemical compound, drug | NADH | Roche (Sigma-Aldrich) | 10107735001 | |
| Chemical compound, drug | Tris base (Amresco) | VWR | 97061–800 | |
| Chemical compound, drug | Boric Acid (ACS grade) | VWR | 97061–980 | |

*Continued on next page*

*Continued*

| Reagent type (species) or resource | Designation | Source or reference | Identifiers | Additional information |
|---|---|---|---|---|
| Chemical compound, drug | EDTA disodium salt dyhydrate | VWR | 97061–018 | |
| Chemical compound, drug | 0.5 M EDTA pH 8 | ThermoFisher | AM9260G | |
| Chemical compound, drug | Dibasic Sodium phosphate | EMD Millipore | SX0715-1 | |
| Chemical compound, drug | Sodium Chloride | EMD Millipore | SX0420-3 | |
| Chemical compound, drug | Potassium Chloride | EMD Millipore | 7300–500 GM | |
| Chemical compound, drug | Sodium Citrate | EMD Millipore | 7810–1 KG | |
| Chemical compound, drug | Sodium Acetate, trihydrate | VWR | MK736406 | |
| Chemical compound, drug | Ficoll 400 | VWR | AAB22095-18 | |
| Chemical compound, drug | Polyvinylpyrrolidone | EMD Millipore | 7220–1 KG | |
| Chemical compound, drug | Diethyl Pyrocarbonate (DEPC) | VWR | AAB22753-14 | |
| Chemical compound, drug | Formamide, deionized | VWR | EM-4610 | |
| Chemical compound, drug | Sodium dodecylsulfate (SDS) | VWR | 97064–470 | |
| Chemical compound, drug | Magnesium chloride hexahydrate | VWR | EM-5980 | |
| Chemical compound, drug | Magnesium sulfate heptahydrate | VWR | EM-MX0070-1 | |
| Chemical compound, drug | Glycerol (ACS grade) | VWR | EMGX0185-5 | |
| Chemical compound, drug | Bovine Serum Albumin (BSA) fraction V | VWR | 101174–932 | |
| Chemical compound, drug | Bromophenol Blue | VWR | EM-BX1410-7 | |
| Chemical compound, drug | Xylene Cyanol | Sigma-Aldrich | X4126-10G | |
| Chemical compound, drug | Amaranth Dye | VWR | 200030–400 | |
| Chemical compound, drug | Temed (JT Baker) | VWR | JT4098-1 | |
| Chemical compound, drug | Ammonium Persulfate | VWR | 97064–594 | |
| Chemical compound, drug | Dithiothreitol (DTT) | Gold Bio | DTT50 | |
| Chemical compound, drug | Glycogen from Oyster (type II) | Sigma-Aldrich | G8751 | |
| Chemical compound, drug | Hydrochloric Acid (ACS plus) | Fisher Scientific | A144-212 | |
| Chemical compound, drug | Ethyl Alcohol | Pharmco-AAPER | 111000200 | |
| Chemical compound, drug | GeneMate LE Quick Dissolve agarose | BioExpress | E-3119–500 | |

*Continued on next page*

*Continued*

| Reagent type (species) or resource | Designation | Source or reference | Identifiers | Additional information |
|---|---|---|---|---|
| Chemical compound, drug | SequaGel sequencing system | National Diagnostics | EC833 | |
| Chemical compound, drug | Nytran SuPerCharge Nylon Membrane | VWR | 10416296 | |
| Chemical compound, drug | SigmaSpin G25 cleanup columns | Sigma-Aldrich | S5059 | |
| Chemical compound, drug | $^{32}$P NAD$^+$ 250 uCi | Perkin Elmer | BLU023X250UC | |
| Chemical compound, drug | γ-$^{32}$P ATP Easy Tide 1 mCi | Perkin Elmer | BLU502Z001MC | |
| Chemical compound, drug | α-$^{32}$P CTP Easy Tide 250 uCi | Perkin Elmer | BLU508H250UC | |
| Chemical compound, drug | α-$^{32}$P GTP Easy Tide 250 uCi | Perkin Elmer | BLU506H250UC | |
| Chemical compound, drug | α-$^{32}$P UTP Easy Tide 250 uCi | Perkin Elmer | BLU507H250UC | |
| Chemical compound, drug | Decade Marker | Thermo Fisher | AM7778 | |
| Chemical compound, drug | TRI Reagent | Molecular Research Center | TR118 | |
| Chemical compound, drug | Acid phenol:chloroform (CHCl$_3$) pH 4.5 | ThermoFisher | AM9720 | |
| Chemical compound, drug | FK866 hydrochloride hyrate | Sigma-Aldrich | F8557 | |
| Software, algorithm | Excel | Microsoft | 365 | |
| Software, algorithm | ImageQuant | GE Healthcare | TL 5.1, TL v8.2 | |
| Software, algorithm | SigmaPlot | Systat Software Inc. | Version 10 | |
| Software, algorithm | Pymol | Schrodinger, LLC | http://www.pymol.org | |
| Software, algorithm | Illustrator | Adobe | Version CS6 | |
| Other | Typhoon RBG Imager | GE Healthcare | | |
| Other | NanoDrop 2000C spectrophotometer | Thermo Fisher | | |
| Other | UV Crosslinker | Fisher Scientific | FB-UVXL-1000 | |
| Other | Hybridization oven 5420 | VWR | 97005–252 | |
| Other | Sequi-Gen GT sequencing systems (21 × 50) (38 × 30) | Bio-Rad | 1653871 and 1653873 | |

## Proteins

*S. cerevisiae* mtRNAP (Rpo41) was prepared from *E. coli* strain BL21(DE3) transformed with pJJ1399 (gift of Judith A. Jaehning) using culture and induction procedures, polyethyleneimine treatment, ammonium sulfate precipitation, Ni-sepharose, DEAE sepharose and Heparin-sepharose chromatography as in (*Tang et al., 2009*). *S. cerevisiae* Mtf1 was prepared from *E. coli* strain BL21(DE3) transformed with pTrcHisC-Mtf1 (*Paratkar and Patel, 2010*) and purified using culture and induction procedures, polyethyleneimine treatment, ammonium sulfate precipitation, and tandem DEAE sepharose and Ni-sepharose chromatography as in (*Tang et al., 2009*).

Human mtRNAP (POLRMT) and TFAM were purified from *E. coli* strain BL21(DE3) transformed with pPROEXHTb-POLRMT(43–1230)–6xHis (*Ramachandran et al., 2017*) or pPROEXHTb-TFAM (43-245)–6xHis (*Ramachandran et al., 2017*), respectively, using culture and induction procedures, polyethyleneimine treatment, ammonium sulfate precipitation, and Ni-sepharose and heparin-sepharose chromatography as in (*Ramachandran et al., 2017*). Human TFB2 was purified from *E. coli* strain ArcticExpress (DE3) (Stratagene) transformed with pT7TEV-HMBP4 (*Yakubovskaya et al.,*

*2014*), using culture and induction procedures, polyethyleneimine treatment, ammonium sulfate precipitation, Ni-sepharose and heparin-sepharose chromatography, and size exclusion chromatography as in (*Yakubovskaya et al., 2014*).

T7 RNAP was prepared from *E. coli* strain BL21 transformed with pAR1219 using culture and inductions procedures, SP-Sephadex, CM-Sephadex and DEAE-Sephacel chromatography as described in (*Jia et al., 1996*).

*E. coli* RNAP core enzyme was prepared from *E. coli* strain NiCo21(DE3) transformed with plasmid pIA900 (*Artsimovitch et al., 2003*) using culture and induction procedures, immobilized-metal-ion affinity chromatography on Ni-NTA agarose, and affinity chromatography on Heparin HP as in (*Artsimovitch et al., 2003*).

*S. cerevisiae* RNA polymerase II core enzyme (gift of Craig Kaplan) was prepared as described in (*Barnes et al., 2015*).

*E. coli* NudC was prepared from *E. coli* strain NiCo21(DE3) transformed with plasmid pET NudC-His (*Bird et al., 2016*) using metal-ion chromatography and size-exclusion chromatography as in (*Cahová et al., 2015*).

RNA 5' pyrophosphohydrolase (RppH) and T4 polynucleotide kinase (PNK) were purchased from New England Biolabs (NEB). FastAP Thermosensitive Alkaline Phosphatase was purchased from Thermo Fisher Scientific. Molar concentrations of purified proteins were determined by light absorbance at 280 nm and the calculated respective molar extinction coefficients.

## Oligodeoxyribonucleotides

Sequences of the oligodeoxyribonucleotides used in this work are provided in Key Resources Table. All oligodeoxyribonucleotides were purchased from Integrated DNA Technologies (IDT) with standard desalting purification unless otherwise specified.

Linear in vitro transcription templates used for transcription assays shown in *Figures 1*, *2* and *3* were generated by mixing complementary equimolar amounts of nontemplate- and template-strand DNA in 10 mM Tris HCl pH 8.0, incubating the mixture at 95°C for 5 min, and cooling the mixture by 0.5°C per minute to 25°C.

Transcription templates used to generate in vitro RNA standards for Northern analysis (*Figures 4* and *5*) were generated by PCR. Reactions contained a mixture of 5 nM template oligo, 0.5 µM forward primer, 0.5 µM reverse primer, and Phusion HF Master Mix (Thermo Scientific). Reaction products were isolated using a Monarch PCR and DNA cleanup kit (NEB).

The radiolabeled 10-nt marker labeled 'M' in gel images shown in *Figures 1*, *4* and *5* was generated using the Decade Marker System (Thermo Fisher Scientific), PNK (NEB) and [γ³²P]-ATP (Perkin Elmer; 6,000 Ci/mmol).

## In vitro transcription assays

Assays performed with *S. cerevisiae* mtRNAP were based on procedures described in (*Deshpande and Patel, 2014*). Assays performed with human mtRNAP were based on procedures described in (*Ramachandran et al., 2017*).

For initial RNA product assays in *Figure 1C*, *Figure 1—figure supplement 1*, and *Figure 3*, 1 µM DNA template, 1 µM *S. cerevisiae* mtRNAP, and 1 µM Mtf1 were incubated at 25°C for 10 min in *Sce*-mtRNAP reaction buffer (50 mM Tris-acetate pH 7.5, 100 mM potassium glutamate, 10 mM magnesium acetate, 0.01% Tween-20, 1 mM DTT, and 5% glycerol). A mixture containing the initiating nucleotide (200 µM ATP, 1 mM NAD⁺, or 1 mM NADH) and extending nucleotide (10 µM of non-radiolabeled GTP plus 6 mCi of [α³²P]-GTP [Perkin Elmer; 3,000 Ci/mmol]) was added, and assays were incubated at 25°C for 30 min. For assays in *Figure 1C*, *Figure 1—figure supplement 1* and radiolabeled initial products were isolated using a Nanosep 3 kDa cutoff spin concentrator (Pall). For assays in *Figure 3B*, reactions were stopped with 10 µl RNA loading dye (95% deionized formamide, 18 mM EDTA, 0.25% SDS, xylene cyanol, bromophenol blue, amaranth) and were analyzed by electrophoresis on 7.5 M urea, 1x TBE, 20% polyacrylamide gels (UreaGel System; National Diagnostics), followed by storage-phosphor imaging (Typhoon 9400 variable-mode imager; GE Life Science).

For the initial RNA product assays in *Figure 1D* and *Figure 1—figure supplement 1*, 1 µM DNA template, 1 µM human mtRNAP, 1 µM TFAM, and 1 µM TFB2M were incubated at 25°C for 10 min

in human-mtRNAP reaction buffer (50 mM Tris-acetate pH 7.5, 50 mM sodium glutamate, 10 mM magnesium acetate, 1 mM DTT, and 0.05% Tween-20). A mixture containing the initiating nucleotide (200 µM ATP, 1 mM NAD$^+$, or 1 mM NADH) and extending nucleotide (10 µM of non-radiolabeled GTP plus 6 mCi of [$\alpha^{32}$P]-GTP at [Perkin Elmer; 3,000 Ci/mmol]) was added, and assays were incubated at 25°C for 60 min. Radiolabeled initial RNA products were isolated using a Nanosep 3 kDa cutoff spin concentrator (Pall).

A portion of the recovered initial RNA products were mixed with either 10 U of RppH or 400 nM NudC and incubated at 37°C for 30 min. Reactions were stopped by addition of 10 µl RNA loading dye. Samples were analyzed by electrophoresis on 7.5 M urea, 1x TBE, 20% polyacrylamide gels (UreaGel System; National Diagnostics), followed by storage-phosphor imaging (Typhoon 9400 variable-mode imager; GE Life Science).

For full-length product assays in *Figure 1C* and *Figure 1—figure supplement 1* (panel B), 1 µM DNA template, 1 µM *S. cerevisiae* mtRNAP, and 1 µM Mtf1 were incubated at 25°C for 10 min in *Sce*-mtRNAP reaction buffer. A mixture containing the initiating nucleotide (1 mM ATP, 1 mM NAD$^+$, or 1 mM NADH for *Figure 1C*; 200 µM non-radiolabeled ATP plus 10 µCi [$\gamma^{32}$P]-ATP [Perkin Elmer; 6,000 Ci/mmol] or 1 mM NAD$^+$ plus 20 µCi [$\alpha^{32}$P]-NAD$^+$ [Perkin Elmer; 800 Ci/mmol] for *Figure 1—figure supplement 1*) and extending nucleotides (200 µM GTP, 200 µM 3'-deoxy-CTP, 20 µM ATP, 200 µM non-radiolabeled UTP, and 6 mCi of [$\alpha^{32}$P]-UTP [Perkin Elmer; 3000 Ci/mmol] for *Figure 1C*; 200 µM GTP, 200 µM 3'-deoxy-CTP, 20 µM ATP, 200 µM UTP for *Figure 1—figure supplement 1*) was added, and assays were incubated at 25°C for 30 min. Reactions were stopped by addition of stop solution (0.6 M Tris HCl pH 8.0, 18 mM EDTA, 0.1 mg/ml glycogen), samples were extracted with acid phenol:chloroform (5:1, pH 4.5; Thermo Fisher Scientific), and RNA products were recovered by ethanol precipitation and resuspended in NudC reaction buffer (10 mM Tris HCl pH 8.0, 50 mM NaCl, 10 mM MgCl$_2$, 1 mM DTT).

For full-length product assays in *Figure 1D* and *Figure 1—figure supplement 1* (panel C), 1 µM DNA template, 1 µM human mtRNAP, 1 µM TFAM, and 1 µM TFB2M were incubated at 25°C for 10 min in human-mtRNAP reaction buffer. A mixture containing the initiating nucleotide (1 mM ATP, 1 mM NAD$^+$, or 1 mM NADH for *Figure 1D*; 200 µM non-radiolabeled ATP plus 10 µCi [$\gamma^{32}$P]-ATP [Perkin Elmer; 6000 Ci/mmol] or 1 mM NAD$^+$ plus 20 µCi [$\alpha^{32}$P]-NAD$^+$ [Perkin Elmer; 800 Ci/mmol] for *Figure 1—figure supplement 1*) and extending nucleotides (200 µM GTP, 20 µM ATP, 200 µM non-radiolabeled UTP, and 6 mCi of [$\alpha^{32}$P]-UTP [Perkin Elmer; 3000 Ci/mmol] for *Figure 1D*; 200 µM GTP, 20 µM ATP, 200 µM UTP for *Figure 1—figure supplement 1*) was added, and assays were incubated at 25°C for 60 min. Reactions were stopped by addition of stop solution, samples were extracted with acid phenol:chloroform (5:1) (pH 4.5; Thermo Fisher Scientific), RNA products were recovered by ethanol precipitation and resuspended in NudC reaction buffer.

Full-length RNA products were incubated at 37°C for 30 min with 400 nM NudC alone (*Figure 1C–D* and *Figure 1—figure supplement 1*), 0.25 U FastAP Thermosensitive Alkaline Phosphatase alone (*Figure 1—figure supplement 1*), or both NudC and FastAP (*Figure 1—figure supplement 1*). Reactions were stopped by addition of an equal volume of RNA loading dye. Samples were analyzed by electrophoresis on 7.5 M urea, 1x TBE, 20% polyacrylamide gels (UreaGel System; National Diagnostics), followed by storage-phosphor imaging (Typhoon 9400 variable-mode imager; GE Life Science).

## Determination of efficiency of NCIN-mediated initiation vs. ATP-mediated initiation, $(k_{cat}/K_M)_{NCIN}$ / $(k_{cat}/K_M)_{ATP}$, in vitro: full-length product assays

For experiments in *Figure 2A–B* and *Figure 2—figure supplement 1*, 1 µM of template DNA, 1 µM of mtRNAP, and 1 µM transcription factor(s) (Mtf1 for *S. cerevisiae* mtRNAP; TFAM and TFB2M for human mtRNAP) were incubated at 25°C for 10 min in *Sce*-mtRNAP or human reaction buffer. A mixture containing 200 µM ATP, 200 µM UTP, 200 µM non-radiolabeled GTP, and 6 mCi [$\alpha^{32}$P]-GTP at 3000 Ci/mmol and NCIN (0, 50, 100, 200, 400, 800, 1600, 3200, 6400 µM) was added, and assays were incubated at 25°C for 30 min. Reactions were stopped by addition of an equal volume of RNA loading dye. Samples were analyzed by electrophoresis on 7.5 M urea, 1x TBE, 20% polyacrylamide gels (UreaGel System; National Diagnostics) supplemented with 0.2% 3-acrylamidophenylboronic acid (Boron Molecular), followed by storage-phosphor imaging (Typhoon 9400 variable-mode imager; GE Life Science).

Bands corresponding to uncapped (pppRNA) and NCIN-capped (NCIN-RNA) full-length products were quantified using ImageQuant software. The ratio of NCIN-RNA to total RNA [NCIN-RNA / (pppRNA + NCIN RNA)] was plotted vs. the relative concentrations of NCIN vs. ATP ([NCIN] / [ATP]) on a semi-log plot (SigmaPlot) and non-linear regression was used to fit the data to the equation: y = (ax) / (b + x); where y is [NCIN-RNA / (pppRNA + NCIN RNA)], x is ([NCIN] / [ATP]), and a and b are regression parameters. The resulting fit yields the value of x for which y = 0.5. The relative efficiency $(k_{cat}/K_M)_{NCIN} / (k_{cat}/K_M)_{ATP}$ is equal to 1/x. Data for determination of relative efficiencies are means of three technical replicates.

## Determination of efficiency of NCIN-mediated initiation vs. ATP-mediated initiation, $(k_{cat}/K_M)_{NCIN} / (k_{cat}/K_M)_{ATP}$, in vitro: initial product assays (Bird et al., 2017)

For experiments in *Figure 3C–D*, and *Figure 3—figure supplement 1*, 1 μM of template DNA, 1 μM of RNAP, and 1 μM transcription factor(s) (Mtf1 for *S. cerevisiae* mtRNAP; TFAM and TFB2M for human mtRNAP; none for T7 RNAP) were incubated at 25°C for 10 min in *Sce*-mtRNAP, human mtRNAP reaction buffer, or T7 RNAP reaction buffer (40 mM Tris HCl pH 7.9, 6 mM MgCl₂, 2 mM DTT, 2 mM Spermidine). A mixture containing 1 mM NCIN, ATP (0, 25, 50, 100, 200, 400, 800, 1600 μM), 20 μM non-radiolabeled GTP, and 6 mCi [$α^{32}$P]-GTP at 3000 Ci/mmol was added, and assays were incubated at 25°C for 30 min. Reactions were stopped by addition of an equal volume of RNA loading dye. Samples were analyzed by electrophoresis on 7.5 M urea, 1x TBE, 20% polyacrylamide gels (UreaGel System; National Diagnostics) supplemented with 0.2% 3-acrylamidophenylboronic acid (Boron Molecular), followed by storage-phosphor imaging (Typhoon 9400 variable-mode imager; GE Life Science).

For experiments in *Figure 2C* and *Figure 2—figure supplements 2*, 200 nM of tailed template and 500 nM RNAP (*S. cerevisiae* mtRNAP, human mtRNAP, *S. cerevisiae* RNAP II, T7 RNAP, or *E. coli* RNAP) were incubated at 25°C for 15 min in reaction buffer containing 10 mM Tris pH 8.0, 50 mM potassium glutamate, 10 mM MgCl₂, 2 mM DTT, and 50 ug/ml BSA. A mixture containing NCIN (1 mM NCIN for mtRNAPs, T7 RNAP, and *E. coli* RNAP; 4 mM for *S. cerevisiae* RNAP II), ATP (0, 25, 50, 100, 200, 400, 800, 1600 μM for mtRNAPs and *S. cerevisiae* RNAP II; 0, 6.25, 12.5, 25, 50, 100, 200, 400 μM for *E. coli* RNAP), 10 μM non-radiolabeled CTP, and 6 mCi [$α^{32}$P]-CTP (Perkin Elmer; 3000 Ci/mmol) was added, and assays were incubated at 25°C for 1 hr. Reactions were stopped by addition of an equal volume of RNA loading dye. Samples were analyzed by electrophoresis on 7.5 M urea, 1x TBE, 20% polyacrylamide gels (UreaGel System; National Diagnostics), followed by storage-phosphor imaging (Typhoon 9400 variable-mode imager; GE Life Science).

Bands corresponding to uncapped (pppApC) and NCIN-capped (NCINpC) initial RNA products were quantified using ImageQuant software. The ratio of NCINpC to total RNA (NCINpC / [pppApC + NCINpC]) was plotted vs. the relative concentrations of NCIN vs. ATP ([NCIN] / [ATP]) on a semi-log plot (SigmaPlot) and non-linear regression was used to fit the data to the equation: y = (ax) / (b + x); where y is [NCINpC / (pppApC + NCINpC)], x is ([NCIN] / [ATP]), and a and b are regression parameters. The resulting fit yields the value of x for which y = 0.5. The relative efficiency $(k_{cat}/K_M)_{NCIN} / (k_{cat}/K_M)_{ATP}$ is equal to 1/x. Data for determination of relative efficiencies are means of three technical replicates.

## Detection and quantitation of NAD⁺- and NADH-capped mitochondrial RNA in vivo: isolation of total cellular RNA from *S. cerevisiae*

For analysis of NAD⁺ and NADH capping during respiration, *S. cerevisiae* strain 246.1.1 [(Tatchell et al., 1981); *MATα ura3 trp1 leu2 his4*; gift of Andrew Vershon, Rutgers University] was grown at 30°C in 25 ml YPEG (24 g Bacto-tryptone, 20 g Bacto-yeast extract, 30 mL ethanol, 3% glycerol per liter) in 125 ml flasks (Bellco) shaken at 220 rpm. When cell density reached an OD600 ~1.8 (approximately 24 hr) the cell suspension was centrifuged to collect cells (5 min, 10,000 g at 4°C), supernatants were removed, and cell pellets were resuspended in 0.8 mL RNA extraction buffer (0.5 mM NaOAc pH 5.5, 10 mM EDTA, 0.5% SDS).

For analysis of NAD⁺ and NADH capping during fermentation, *S. cerevisiae* strain 246.1.1 was grown at 30°C in 100 ml YPD [24 g Bacto-tryptone, 20 g Bacto-yeast extract, 2% (w/v) glucose per liter] in 125 ml flasks (Bellco) with airlocks to prevent oxygenation without shaking for 42 hr. The cell

suspension was centrifuged to collect cells (5 min, 10,000 g at 4°C), supernatants were removed, and cell pellets were resuspended in 0.8 mL RNA extraction buffer (0.5 mM NaOAc pH 5.5, 10 mM EDTA, 0.5% SDS).

To extract RNA, an equal volume of acid phenol:chloroform (5:1, pH 4.5; Thermo Fisher Scientific) was added to cells in resuspension buffer and mixed by vortexing. Samples were incubated at 65°C for 5 min, −80°C for 5 min, then centrifuged (15 min, 21,000 g, 4°C) to separate the aqueous and organic phases. The aqueous phase was collected and acid phenol:chloroform extraction was performed two more times on this solution. RNA transcripts were recovered by ethanol precipitation and resuspended in RNase free $H_2O$.

## Detection and quantitation of NAD$^+$- and NADH-capped mitochondrial RNA in vivo: isolation of total cellular RNA from human cells

Human embryonic kidney HEK293T cells (obtained from ATCC, tested negative for mycoplasma) were maintained under 5% $CO_2$ at 37°C in DMEM medium (Thermo Fisher Scientific) supplemented with 10% fetal bovine serum (Atlanta Biologicals), 100 units/ml penicillin, and 100 µg/ml streptomycin. HEK293T cells were seeded in 100 mm tissue-culture treated plates and grown for 72 hr at 37°C or seeded in 100 mm tissue-culture treated plates, grown for 24 hr at 37°C, treated with 5 nM FK866 (Sigma Aldrich), and grown for an additional 48 hr at 37°C. Total cellular RNA was isolated with TRIzol Reagent according to the manufacture's protocol (Thermo Fisher Scientific).

## Detection and quantitation of NAD$^+$- and NADH-capped mitochondrial RNA in vivo: DNAzyme cleavage

For analysis of NCIN capping of *S. cerevisiae* mitochondrial RNA, 40 µg of total cellular RNA was mixed with 1 µM DNAzyme (JB557 for *S. cerevisiae* COX2 RNA; JB526 for *S. cerevisiae* 21S RNA) in buffer containing 10 mM Tris pH 8.0, 50 mM NaCl, 2 mM DTT, and 10 mM $MgCl_2$ (total volume 50 µl). When present, NudC was added to 400 nM. Reactions were incubated for 60 min at 37°C and 100 µl of stop solution and 500 µl ethanol was added. Samples were centrifuged (30 min, 21,000 g, 4°C), the supernatant removed, and the pellet resuspended in RNA loading dye.

For analysis of NCIN capping of human mitochondrial RNA, 40 µg of total cellular RNA was mixed with 1 µM DNAzyme (JB559 for human LSP-generated RNA) in buffer containing 10 mM Tris pH 8.0, 50 mM NaCl, 2 mM DTT (total volume 50 µl). Samples were heated to 85°C for 5 min, cooled to 37°C. $MgCl_2$ was added to a final concentration of 10 mM and, when present, NudC was added to 400 nM. Reactions were incubated for 60 min at 37°C and 100 µl of stop solution and 500 µl ethanol was added. Samples were centrifuged (30 min, 21,000 g, 4°C), the supernatant removed, and the pellet resuspended in RNA loading dye.

To prepare synthetic RNA standards non-radiolabeled full-length RNA products were generated by in vitro transcription reactions containing 1 µM of template DNA, 1 µM of mtRNAP, 1 µM transcription factor(s), 1 mM initiating nucleotide (ATP, NAD$^+$ or NADH), 200 µM GTP, 200 µM UTP, 200 µM CTP, and 20 µM ATP. Reactions were incubated for 60 min at 25°C, stopped by addition of stop solution, extracted with acid phenol:chloroform (5:1, pH 4.5; Thermo Fisher Scientific) and ethanol precipitated. Full-length RNAs were resuspended in buffer containing 10 mM Tris pH 8.0, 50 mM NaCl, 2 mM DTT, and 10 mM $MgCl_2$ and treated with DNAzyme as described above.

## Detection and quantitation of NAD$^+$- and NADH-capped mitochondrial RNA in vivo: hybridization with a radiolabeled oligodeoxyribonucleotide probe

NCIN capping of DNAzyme-generated subfragments of mitochondrial RNA were analyzed by a procedure consisting of: (i) electrophoresis on 7.5 M urea, 1x TBE, 10% polyacrylamide gels supplemented with 0.2% 3-acrylamidophenylboronic acid (Boron Molecular); (ii) transfer of nucleic acids to a Nytran supercharge nylon membrane (GE Healthcare Life Sciences) using a semidry transfer apparatus (Bio-Rad); (iii) immobilization of nucleic acids by UV crosslinking; (iv) incubation with a $^{32}$P-labelled oligodeoxyribonucleotide probe complementary to the 5'-end containing subfragments of target RNAs (JB555, COX2 RNA; JB525, 21S RNA; JB515, LSP-derived RNA; $^{32}$P-labelled using T4 polynucleotide kinase and [γ$^{32}$P]-ATP [Perkin Elmer]); (v) high-stringency washing, procedures as in

(*Goldman et al., 2015*); and (vi) storage-phosphor imaging (Typhoon 9400 variable-mode imager; GE Life Science).

Bands corresponding to uncapped and NCIN-capped DNAzyme-generated subfragments were quantified using ImageQuant software. The percentages of uncapped RNA (5'-ppp), $NAD^+$-capped RNA (5'-$NAD^+$), or NADH-capped RNA (5'-NADH) to total RNA were determined from three biological replicates.

## Determination of NAD(H) levels in human HEK293T cells

Cells were lysed in 400 µl of NAD/H Extraction Buffer (NAD/H Quantitation Kit, Sigma-Aldrich) and total proteins were extracted with two freeze-thaw cycles (20 min on dry ice, 10 min at RT). Lysates were centrifuged (10 min, 13,000 g, 4°C), the supernatant was isolated, and the protein concentration was determined. NAD(H) was measured using a NAD/H Quantitation Kit (Sigma) from a volume of supernatant containing 10 µg protein.

Molecules of NAD(H) per cell were calculated using a value of 300 pg total protein per cell. The cellular concentration of NAD(H) was then estimated by using a cellular volume of ~1200 $\mu m^3$ for HEK293T cells and assuming a homogenous distribution of NAD(H) in the cell. The volume of HEK293T cells was calculated using a value of 13 µm for the diameter of HEK293T cells, assuming trypsinized cells adopt a spherical shape. NAD/H concentrations were determined from three biological replicates.

# Additional information

## Funding

| Funder | Grant reference number | Author |
| --- | --- | --- |
| American Heart Association | 16PRE30400001 | Urmimala Basu |
| National Institutes of Health | NS021328 | Douglas C Wallace |
| National Institutes of Health | MH108592 | Douglas C Wallace |
| National Institutes of Health | OD010944 | Douglas C Wallace |
| U.S. Department of Defense | W81XWH-16-1-0401 | Douglas C Wallace |
| National Institutes of Health | GM126488 | Megerditch Kiledjian |
| National Institutes of Health | GM104231 | Dmitry Temiakov |
| National Institutes of Health | GM118086 | Smita S Patel |
| National Institutes of Health | GM041376 | Richard H Ebright |
| National Institutes of Health | GM118059 | Bryce E Nickels |

The funders had no role in study design, data collection and interpretation, or the decision to submit the work for publication.

## Author contributions

Jeremy G Bird, David Kuster, Formal analysis, Investigation, Visualization, Methodology, Writing—review and editing; Urmimala Basu, Formal analysis, Funding acquisition, Investigation, Visualization, Methodology, Writing—review and editing; Aparna Ramachandran, Investigation, Methodology, Writing—review and editing; Ewa Grudzien-Nogalska, Atif Towheed, Investigation; Douglas C Wallace, Supervision, Funding acquisition; Megerditch Kiledjian, Supervision, Funding acquisition, Writing—review and editing; Dmitry Temiakov, Conceptualization, Funding acquisition, Methodology, Writing—review and editing; Smita S Patel, Conceptualization, Supervision, Funding acquisition, Visualization, Project administration, Writing—review and editing; Richard H Ebright, Bryce E Nickels, Conceptualization, Supervision, Funding acquisition, Visualization, Writing—original draft, Project administration

Author ORCIDs
David Kuster http://orcid.org/0000-0002-8157-9223
Richard H Ebright http://orcid.org/0000-0001-8915-7140
Bryce E Nickels http://orcid.org/0000-0001-7449-8831

Decision letter and Author response
Decision letter https://doi.org/10.7554/eLife.42179.022
Author response https://doi.org/10.7554/eLife.42179.023

## Additional files

### Supplementary files
• Transparent reporting form
DOI: https://doi.org/10.7554/eLife.42179.020

### Data availability
All data generated or analysed during this study are included in the manuscript.

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
