## [Decision Letter]

Thank you for submitting your article "Highly efficient 5' capping of mitochondrial RNA with NAD+ and NADH by yeast and human mitochondrial RNA polymerase" for consideration by *eLife*. Your article has been reviewed by three peer reviewers, one of whom is a member of our Board of Reviewing Editors, and the evaluation has been overseen by James Manley as the Senior Editor. The following individual involved in review of your submission has agreed to reveal their identity: Ann Hochschild (Reviewer #2).

The reviewers have discussed the reviews with one another and the Reviewing Editor has drafted this decision to help you prepare a revised submission.

Summary:

This paper uses a combination of in-depth biochemical analysis and a novel in vivo approach to provide important new insights into NAD/NADH capping of eukaryotic mitochondrial transcripts and of this capping process in general. Using purified RNA polymerases from yeast or human mitochondria (mt), as well yeast nuclear, *E. coli* and T7 RNA Pol enzymes, they show by careful enzyme kinetics that the mt enzymes are considerably more efficient than the yeast and *E. coli* enzymes at NAD+ capping in vitro. They further show that the mt enzymes and T7 Pol, all single polypeptide polymerases, exhibit the same promoter sequence specificity for the reaction demonstrated previously for the *E. coli* multi-subunit enzyme, requiring a T on the template strand for the TSS (+1 position) and a purine:pyrimidine base pair on the non-template:template strands at the -1 position. They introduce a significant advance in the techniques for quantifying the occurrence of NAD+ and NADH, vs uncapped mt transcripts produced in vivo, and show that the proportions of NAD+/NADH vs. uncapped transcripts for two yeast and one human mt transcripts are much higher than observed previously for bacterial transcripts and yeast cytoplasmic mRNAs. Finally, they show that the ratio of NAD+ vs NADH capped mRNAs for the two yeast mt mRNAs vary in response to altered growth conditions expected to change the NAD+/NADH ratio in cells, providing evidence that mt RNAP could be utilized as a sensor of energy metabolism that would generate altered ratios of NAD+ vs NADH-capped mt mRNAs. In view of their findings that a large fraction of the mt transcripts harbor one or the other cap, this has the potential of altering mt gene expression through possible influences of these caps on mRNA translation or turnover.

Although NAD+ capped yeast mt mRNAs had been detected previously, this work is significant in providing direct evidence that mt RNAPs carry out this reaction and utilize the same promoter sequences found for multi-subunit RNAPs to direct capping; and they also function much more efficiently than do the latter. The results also provide evidence that the capped mt transcripts detected in cells represent capping during transcription initiation vs. capping of degradation products. By measuring the precise proportions of NAD+-capped, NADH-capped, and uncapped mt mRNAs in cells-which is much higher than seen previously for bacterial or yeast cytoplasmic mRNAs-they show that the ratio of NAD+ to NADH-capped mRNAs varies with the energy status of yeast cells.

Essential revisions:

It is requested that you discuss how the relative levels of NAD/NADH capping of mitochondrial versus cytoplasmic mRNAs could be affected by different concentrations of NAD/NADH in the mitochondria versus eukaryotic nuclei or *E. coli* cells in addition to the different capping efficiencies of the respective RNA polymerases, and cite any information available about these concentrations.

It is requested that you provide a much better explanation of how the "fork-junction" template works and how it bypasses the requirement for sequence-specific RNAP-DNA interactions and transcription-initiation factor-DNA interactions for transcription initiation, as neither the text nor cartoon were found to be adequate.

It is requested that you state more explicitly how the conclusion that "mitochondrial RNAs undergo NAD+ capping at 5' ends generated by transcription initiation (as opposed 5' ends generated by RNA processing" follows from the data provided.

It is requested that you discuss what is known about whether the capping ratio affects the amount of transcript, the stability of the transcript, or the efficiency with which it is translated.

---

## [Author Response]

Essential revisions:It is requested that you discuss how the relative levels of NAD/NADH capping of mitochondrial versus cytoplasmic mRNAs could be affected by different concentrations of NAD/NADH in the mitochondria versus eukaryotic nuclei or E. coli cells in addition to the different capping efficiencies of the respective RNA polymerases, and cite any information available about these concentrations.

We have modified the text in the second paragraph of the Discussion to address this point. The revised text is:

“First, mtRNAPs are substantially more efficient at NAD^+^ and NADH capping than bacterial and eukaryotic nuclear RNAPs (Figure 2C). Second, levels of NAD^+^ and NADH relative to ATP in mitochondria are substantially higher than levels of NAD^+^ and NADH relative to ATP in bacteria and eukaryotic nuclei (Cambronne et al., 2016; W. W. Chenet al., 2016; Parket al., 2016).”

It is requested that you provide a much better explanation of how the "fork-junction" template works and how it bypasses the requirement for sequence-specific RNAP-DNA interactions and transcription-initiation factor-DNA interactions for transcription initiation, as neither the text nor cartoon were found to be adequate.

We have revised the text to indicate that the template used for these assays is a “tailed” template and provided references establishing use of a tailed template to bypass the requirement for sequence-specific RNAP-DNA interactions and transcription-initiation factor-DNA interactions. The revised text is:

“…we performed transcription assays using a “tailed” template (Figure 2C, top) that bypasses the requirement for sequence-specific RNAP-DNA interactions and transcription-initiation factor-DNA interactions for transcription initiation (Dedrick and Chamberlin, 1985; Kadesch and Chamberlin, 1982).”

It is requested that you state more explicitly how the conclusion that "mitochondrial RNAs undergo NAD+ capping at 5' ends generated by transcription initiation (as opposed 5' ends generated by RNA processing" follows from the data provided.

We have revised the text to clarify this point by adding the following sentence to the Results section:

“Because the hybridization probe detects RNA fragments that contain 5' ends generated by transcription initiation (red sub-fragment depicted in Figure 4A), the detected NAD^+^ and NADH caps are concluded to be at 5' ends generated by transcription initiation, as opposed to 5' ends generated by RNA processing.”

It is requested that you discuss what is known about whether the capping ratio affects the amount of transcript, the stability of the transcript, or the efficiency with which it is translated.

We have modified the text at the beginning of the second paragraph of the Discussion. The revised text is:

“We and others previously have shown that NCIN capping by cellular RNAPs has functional consequences, including modulating RNA stability and modulating RNA translatability.”